# Assessment of Cementitious Composites for High-Temperature Geothermal Wells

**DOI:** 10.3390/ma17061320

**Published:** 2024-03-13

**Authors:** Tatiana Pyatina, Toshifumi Sugama, Al Moghadam, Marcel Naumann, Ragnhild Skorpa, Blandine Feneuil, Vincent Soustelle, Rune Godøy

**Affiliations:** 1Brookhaven National Laboratory, Upton, NY 11973, USA; sugama@bnl.gov; 2TNO Geoscience and Technology, Princetonlaan 6, 3584 CB Utrecht, The Netherlands; al.moghadam@tno.nl (A.M.); vincent.soustelle@tno.nl (V.S.); 3Equinor ASA, Sandslivegen 90, 5254 Sandsli, Norway; mnau@equinor.com; 4SINTEF AS, S. P. Andersens veg, 15B, 7031 Trondheim, Norway; ragnhild.skorpa@sintef.no (R.S.);; 5Equinor ASA, Forusbeen 50, Forus East, 4035 Stavanger, Norway; rugo@equinor.com

**Keywords:** high-temperature OPC, geothermal cement, calcium–aluminate well cement, function cement tests, high-temperature cement CO_2_ resistance, finite element modeling, calcium-free well cement

## Abstract

High-temperature (HT) geothermal wells can provide green power 24 hours a day, 7 days a week. Under harsh environmental and operational conditions, the long-term durability requirements of such wells require special cementitious composites for well construction. This paper reports a comprehensive assessment of geothermal cement composites in cyclic pressure function laboratory tests and field exposures in an HT geothermal well (300–350 °C), as well as a numerical model to complement the experimental results. Performances of calcium–aluminate cement (CAC)-based composites and calcium-free cement were compared against the reference ordinary Portland cement (OPC)/silica blend. The stability and degradation of the tested materials were characterized by crystalline composition, thermo-gravimetric and elemental analyses, morphological studies, water-fillable porosity, and mechanical property measurements. All CAC-based formulations outperformed the reference blend both in the function and exposure tests. The reference OPC/silica lost its mechanical properties during the 9-month well exposure through extensive HT carbonation, while the properties of the CAC-based blends improved over that period. The Modified Cam-Clay (MCC) plasticity parameters of several HT cement formulations were extracted from triaxial and Brazilian tests and verified against the experimental results of function cyclic tests. These parameters can be used in well integrity models to predict the field-scale behavior of the cement sheath under geothermal well conditions.

## 1. Introduction

High-temperature (HT) geothermal wells may provide green energy available 24/7. Durable and sustainable materials are required for the construction and operation of HT wells. The material requirements for subterranean wells for Enhanced Geothermal Systems (EGS) that involve geoformation hydraulic stimulation for the efficient heat recovery of the earth are especially stringent. These wells have larger diameters than oil and gas wells and are constructed and operated under higher temperatures, more chemically aggressive environments, and repeated thermo-mechanical stresses. These wells are expected to have a long service life for a power plant installation, so long-term material durability is important. 

The power output of geothermal power plants can be further increased with super-hot and supercritical wells that offer significant economic benefits. The energy production from a single well with a temperature above 400 °C can be 10 times higher than that from a regular geothermal well and 4–5 times higher than from a well in shale gas fields [1,2,3]. However, targeting higher temperatures imposes new challenges on well construction and the materials used in such wells. Material damage and degradation reactions can dramatically accelerate under HT conditions.

Geothermal environments are often very rich in acidic gases (CO_2_, H_2_S) and dissolved solids, making it more difficult for cements to survive than oil and gas wells [4]. The decalcification of ordinary Portland cement (OPC) with a loss of its mechanical properties and an increase in permeability upon long-term exposure to acidic fluids has been known for a long time [5]. Extensive research has focused on the safety of underground CO_2_ storage and the appropriateness of the regulations concerning the length of the storage well plugs [6,7,8]. The general conclusion was that under the conditions of the storage wells, the kinetics of cement degradation are very slow due to the formation of a carbonated cement layer that decreases cement permeability and increases its brittleness. This was confirmed by the natural CO_2_ producer [9] and by pilot storage wells [10] that proved to operate without issues for extended periods of time. Cement sample well exposure tests also confirmed the OPC/silica HT blend to be stable for extended periods of time at temperatures below 200 °C [11]. In this study, formulations with higher silica content were shown to be preferential for good mechanical performance and lower porosity of cement after up to a year of field exposure. However, all these studies applicable to CO_2_ storage wells address conditions that are far from those found in geothermal wells, with significantly higher temperatures, more complex brine compositions, and shock conditions that can damage cement and accelerate its chemical degradation through cracks and fissures.

Previous laboratory work demonstrated poor acid resistance of OPC-based formulations at HT compared against that of calcium–aluminate-based cements [12]. To resist the commonly high CO_2_ content of geothermal wells, Brookhaven National Laboratory (BNL) developed a chemical calcium phosphate cement that can mineralize CO_2_ into a stable carbonated apatite phase [13]. Calcium phosphate cement was also shown to self-heal under HT geothermal conditions for controlled crack width [14]. To resist thermal shocks common to HT geothermal wells, Thermal Shock Resistant Cement (TSRC) was developed and shown to possess outstanding acid resistance and self-healing properties [12,15,16,17]. Preliminary evaluations of these and other cement formulations under supercritical conditions showed persistence or improvement of their properties after up to 30 days at 400 °C and 25 MPa [18,19,20]. However, for many advanced cementitious blends, their performance under the conditions of a real geothermal well over extended periods of time remains to be seen. Laboratory evaluations of most well cements are conducted after their hydrothermal synthesis at HT and high pressure (HP), followed by room-temperature analyses. Reproduction of geothermal well conditions and long-term tests are problematic in laboratory environments. Field exposure of experimental cements in a real deep geothermal well is an attractive alternative to laboratory testing.

Nevertheless, advanced laboratory testing can provide valuable complimentary information on blend performance under well conditions. To assess the isolation provided by cement, pressure testing is most commonly performed after every surface or intermediate-casing cementing job. During the testing, the internal casing pressure is increased to exceed the pressure that will be applied during the next drilling phase [21]. Similarly, casing pressure increases in geothermal wells during hot fluid production. Pressure testing can be used in laboratory setups to compare the performance of different cement formulations in a scaled-down well configuration [22]. In these tests, the development of cracks in the cement sheath between the casing and a rock and cement-casing or cement-formation bond failure are monitored by 3D imaging. Earlier, such function tests were used to show the effects of poor casing centering in a borehole and poor mud removal from the rock surface on cement strength and failure modes [23,24]. Function pressure tests were also used to evaluate cement performance with different types of rocks, where stiff rock surrounding the cement sheath resulted in a brittle fracture of the samples, appearing at a higher load than for a soft rock, where the cracks grew progressively [23]. Most importantly, function tests have been used to compare different cement formulations, for instance in [25], where a CAC-based formulation failed under a higher load than an OPC-based cement after curing at 110 °C.

In addition to laboratory testing mimicking well conditions and well exposures, hydro-thermo-mechanical modeling of the nearby wellbore region has been used to assess well integrity issues [26,27,28]. Calibration of such models with lab experiments is necessary to extend the results of lab experiments to field conditions [29]. Modified Cam-Clay (MCC) plasticity models have been proposed as suitable for predicting cement behavior [30,31]. Such models can predict the strain softening and hardening that are observed in cement and the irreversible compaction of cement during casing pressurization.

Among the difficulties of evaluating and modeling the behavior of well cement is the fact that cementitious materials are exposed to a wide range of temperatures and must have a prolonged life. The solidification reactions are initiated above the ground at ambient temperatures and completed in a well at elevated temperatures. The well temperature varies not only with depth but, in the short run, with time, when the well returns to a higher static temperature after the initial cooling with circulating fluids. The short-term cement sheath survival under aggressive environments generally evaluated in laboratory tests does not guarantee long-term integrity.

The present paper addresses the knowledge gap in the performance of various cementitious formulations under a wide range of conditions relevant for HT geothermal wells, reporting both laboratory and long-term cement field exposure results as well as modeling cement sheath behavior under conditions of thermo-mechanical stresses. The short-term performance of three cement formulations with different characteristic failure behaviors (OPC/silica reference; more ductile, calcium–aluminate cement (CAC)/Silica; and more brittle calcium phosphate (CAP) composites) is evaluated in pressure function tests at a lower temperature (110 °C). Triaxial tests conducted on these formulations are used to extract the MCC model parameters. The parameters are then used to model the function tests and verify the results of the numerical model. The long-term survival of these and other HTHP formulations is evaluated under the HT geothermal well environments in sample exposure tests for three and nine months in a Newberry well in Oregon, US, at temperatures above 300 °C. The well-exposed samples are analyzed for changes in their mechanical properties, porosity, and phase compositions to understand the stability and degradation of cements with different chemistries in an aggressive hot well environment. The paper provides the results of short-term laboratory studies along with the long-term field performance of various cementitious composites.

## 2. Materials and Methods

### 2.1. Materials

Calcium–aluminate cements (CACs), Secar #80, Secar #71, Secar #50, and class G OPC were used in this study. All CACs were supplied by Imerys, Kings Mountain, NC, USA, while Lafarge, North America, New Heaven, CT, USA, provided the OPC, class G, and well cement. The X-ray powder diffraction (XRD) data showed that the crystalline compounds of #80 CAC were the following three principal phases: calcium monoaluminate (CaO·Al_2_O_3_, CA), calcium dialuminate (CaO·2Al_2_O_3_, CA_2_), and corundum (α-Al_2_O_3_); #50 CAC had CA as its dominant phase, coexisting with gehlenite [Ca_2_Al(Al,Si)_2_O_7_] and corundum as the secondary components. The class G cement consisted of hatrurite (3CaO·SiO_2_) as a major phase and brownmillerite (4CaO·Al_2_O_3_.Fe_2_O_3_), basanite (CaSO_4_·1/2H_2_O), and periclase (MgO) as minor phases for the former cement. Among the cement-forming constituents, SMS (Na_2_SiO_3_), an alkali-activating powder of 93% purity with particle sizes of 0.23 to 0.85 mm, trade named “MetsoBeads 2048”, was supplied by the PQ Corporation. It had a 50.5/46.6 Na_2_O/SiO_2_ weight ratio. Sodium hexa-meta-phosphate (SHMP) [(NaPO_3_)_6_, 60–70% P_2_O_5_] with 200 mesh granular obtained from Sigma-Aldrich, St. Louis, MO, USA, was used as a cement-building component of calcium phosphate cements. Silica flour was supplied by Cudd Energy Services. The metakaolin (Al_4_Si_2_O_10_) was obtained from Imerys. FAF was supplied by Boral Material Technologies, San Antonio, TX, USA. The XRD analysis of FAF showed that it included three major crystalline phases: quartz (SiO_2_), mullite (3Al_2_O_3_.2SiO_2_), and hematite (Fe_2_O_3_). Aluminum hydroxide, an EMPLURA^®^ hydragillite powder with a bulk density of ~90 g/100 mL and particle size < 150 µm for 90% of the material, was obtained from Sigma Aldrich, St. Louis, MO, USA. Zirconium (IV) hydroxide as hydrous zirconium oxide, ZrO_2_.nH_2_O (Zr), was also obtained from Sigma Aldrich.

All formulations exposed in the Newberry well for 9 months were modified with 5% by weight of dry blend carbon microfibers (CMF, AGM-94) derived from a polyacrylonitrile precursor supplied by Asbury Graphite Mills, Inc., Asbury, NJ, USA. They were 7–9 microns in diameter and 100–200 microns in length. These fibers are stable at temperatures up to 600 °C. Earlier application of these fibers was shown to improve the ductility and cement–carbon steel bond of HT cement samples, which commonly decrease with longer HT curing times in laboratory tests [32]. They were used to improve the mechanical properties of the tested composites and to confirm their long-term chemical and thermal stability in field tests.

### 2.2. Sample Preparation and Exposure Conditions

#### 2.2.1. Cement Formulations and Sample Preparation

The formulations of cement samples evaluated in this work are presented in Table 1. The compositions of the dry blends are provided as the mass percent of the dry blend components, followed by the mass percent of the activator. For example, CAC#71/FAF/SHMP (70/30, SHMP at 6%) means that this dry blend contained 70 wt.% of CAC#71 and 30 wt.% of FAF. Six percent of SHMP activator by the total weight of CAC#71/FAF was added to make the final blend. The dry blends were prepared by shaking all the dry components for 3 min by hand. Water was then added to dry blends at the amount that allowed the slurries to have equal self-leveling. The water content varied between 0.312/0.4 (for CAP#71/Silica/MK without/with CMF, respectively) and 0.432/0.481 (for #71/Silica/MK without/with CMF, respectively). The slurries were placed into molds (20 × 40 mm) and left at room temperature for 12 h. Then samples were demolded and exposed to an 85 °C environment with a relative humidity of 99.9% for another 12 h. Finally, they were autoclaved at 300 °C for 12 more hours in a non-stirred Parr Reactor 4622 before shipment to the wellsite for exposure. The NAS-M1 formulation was prepared in glass tubes (18 × 150 mm) without demolding, following the temperature regime for all other samples. After 1 day of 300 °C autoclaving, each tube was cut into 3 cylinders of ~40 mm each, and the solidified cement was removed from the glass.

To prepare samples for function tests, dry blend components of #80/Silica and CAP#71/Silica were blended in a Dry Powder Mixer Blending Machine V10 from iPharMachine for 12 h. The blends were then mixed with water (water-to-blend (W/B) ratio of 0.372 and 0.32, respectively) in an Ofite Constant Speed Blender following the API procedure: 15 s at 4000 rpm, followed by 30 s at 12,000 rpm. To prepare the CSH-60/40 slurry, OPC and 45-microns silica flour were first mixed with a spatula to obtain a homogeneous powder, then mixed with water (W/B = 0.44) following the API procedure.

#### 2.2.2. Exposure Tools

Exposure tools were fabricated from stainless steel to host cylindrical samples (Figure 1). They had open slots for sample exposure to the well environment and side plates for centralized placement in the well. Three baskets could be fitted on top of each other and removed together or separately from the well.

The baskets were loaded with samples precured for 1 day at 300 °C and deployed in the well on a wireline, then detached from the wireline and left at the bottom of the well. Since bottom hole cement in geothermal wells would experience repeated thermal shock conditions, no special precautions were taken for samples re-equilibrating with the changing environment during their pull up to the surface. The tested formulations also did not include any foam or other lightweight cement that could be sensitive to the relatively rapid pressure decrease. The samples recovered after both exposures of 3 and 9 months did not have any visible damage.

#### 2.2.3. Exposure Conditions

According to earlier well service data, the bottom hole temperature of the well where the sample baskets were released was ~325 °C (communications with AltaRock). Information on the pressure at the bottom of the well was not available; the pressure at the depth of about 1.6 km was nearly 14 MPa. If a constant pressure gradient is assumed between 1.6 and 3.0 km, the bottom hole pressure can be estimated to be ~26 MPa.

Geochemical analysis from flow testing of this well by Geologica performed in 2008 showed that a non-condensable gas, identified as being >99% CO_2_, was coming from a geological source (hydrothermal or magmatic). The total carbonate concentration measured at 2 different locations in well fluids was 296 and 1930 mg/kg. According to that information, supercritical CO_2_ could be expected at the bottom of the well.

### 2.3. Sample Analyses

#### 2.3.1. Physical, Structural, and Mineralogical Analyses

The percent of water-fillable porosity of the well-exposed samples was determined using the following formula: (W_wet_ − W_dry_)/V × 100, where W_wet_ is the weight of water-saturated samples and W_dry_ is the weight of a sample dried for at least 4 days in a vacuum oven at 60 °C until the weight of the sample was constant, and V is the volume of the sample [33].

JEOL 7600 F (Pleasanton, CA, USA) scanning electron microscope image analysis coupled with energy dispersive X-ray (EDX) elemental composition measurements on freshly broken surfaces was employed for morphological analyses and phase identifications.

TGA/DTA (heating rate of 20 °C/min in a N_2_ flow, ~10 mg sample weight, model Q50, TA Instruments, New Castle, DE, USA) and X-ray diffraction measurements (40 kV, 40 mA copper anode X-ray tube, Rigaku Smartlab, Cedar Park, TX, USA) were used for sample characterization. The PDF-4/Minerals 2023 database of the International Center for Diffraction Data (ICDD) was used for the analysis of XRD patterns. The background was subtracted from all the XRD patterns before their analyses.

#### 2.3.2. Mechanical Tests

The uniaxial compressive strength, Young’s modulus, and compressive toughness were determined using the Electromechanical Instron System Model 5967 (Norwood, MA, USA). The measurements were performed on unconfined, dry samples after the water-fillable porosity determinations. The instrument had a 30 kN load capacity, and the measurements were performed at a 1.25 cm/min loading rate. The measurements were used to obtain the comparative performance of different formulations without focusing on absolute values. The mechanical properties and percent of water-fillable porosity of the reference samples after the 300 °C autoclaving were measured in the same manner. The compressive toughness was computed from the area under the compressive stress–strain curve.

Triaxial tests to determine the deformation and strength properties of the samples were performed using a triaxial cell-type MTS Model 815 Rock Mechanics System (Eden Prairie, MN, USA). Drained tests at room temperature on water-saturated samples at 2.5 MPa pore pressure were conducted at various effective confining stresses to establish input for the constitutive model (Section 2.5). The test procedure mostly followed the suggestions for triaxial testing by ISRM [34]. Sample end surfaces were ground parallel according to ASTM and ISRM standards [35]. Initial full-pore fluid saturation was achieved by vacuum saturation. The otherwise occurring impact of varying degrees of saturation on mechanical properties is well documented in the literature (e.g., [36]).

An axial strain rate below 1 × 10^−6^ 1/s was applied to ensure no internal pore pressure built up. All triaxial tests were conducted in the following manner:Hydrostatic loading to a predefined confinement.Triaxial compression in axial strain control to an axial stress of about 50% of the expected peak stress.Triaxial unloading to initial stress conditions at the same rate.Triaxial compression (re-loading) at the same constant rate until reaching a constant post-failure plateau.

Strains were measured by clip-on extensometers in an axial direction and by a chain extensometer in a circumferential direction (Figure 2). The volumetric strain was derived from those two measurements. Axial stress is recorded with an internal load cell and confining and pore pressure with pressure gauges. The mechanical loading plates of the setup also contain ultrasonic transducers to measure compressional and shear wave velocity at the same time. Those results will be presented in a separate paper.

The triaxial compression tests provided access to the following mechanical parameters as a function of stress state (confinement and shear load):Deformability (e.g., Young’s modulus, Poisson’s ratio, and shear and bulk modulus)Strength (e.g., shear and residual strength, cohesion, and friction angle)Ultrasonic velocity (compressional and shear waves propagating in an axial sample direction)Permeability (measured in steady-state conditions by applying a differential pore pressure and recording the associated fluid flow rate)Porosity (derived from volumetric strain changes)

The tensile strength of the materials was determined by the so-called Brazilian test [37]. Samples with a length-to-diameter ratio of 1:2 were placed sideways between special loading jaws until failure. We followed the ISRM-recommended method presented by [38]. Tests were performed on dry samples in axial displacement control with a loading rate of about 5–10 N/s, depending on the stiffness of the material.

### 2.4. Function Tests

The setup and testing procedure of the function tests have been described in [22] and later used in several studies [23,24,25,39,40]. These cyclic pressure experiments were performed on a small-scale wellbore section of cement sheath between rock and casing. The setup used for functional tests is illustrated in Figure 3.

The carbon-steel casing was 25 cm long, 2 mm thick, and had a 60.3 mm external diameter. The rock used in this work was a hollow cylindrical Berea sandstone (length 20 cm, external diameter 15 cm, and internal diameter 76 mm). The unconfined compressive strength of the rock was about 49 MPa, and the Young’s modulus was ~13 GPa. During sample preparation, the water-saturated rock was placed around the casing in an aluminum cell, and cement slurry was poured into the 8 mm gap between the casing and the rock. The cement was cured at 110 °C for 20 days by placing the cell on a copper plate on a heating platform. A copper rod attached to the copper plate was placed inside the casing to facilitate heating of the sample radially from inside the casing. After cement curing, the copper rod was removed from inside the casing and replaced by the pressure shaft, enabling pressure cycling tests of the sample.

Prior to the function tests, the initial condition of the cement sheath was mapped by a CT scan of the sample, resulting in a 3D visualization. The sample was then pressurized to 15 MPa for 1 min, and a CT scan of the sample was performed during the last 10 s before the pressure was released. The sample was then re-pressurized to 16 MPa for 1 min, and a CT scan of the sample was performed during the last 10 s before the pressure was again released. The procedure was continued, with a 1 MPa increase in pressure at each step, until a complete failure of the sample was observed. A detailed description of the procedure and the following analysis can be found in [41].

The obtained 3D image was composed of 220 horizontal slices (about one every mm), with the spatial resolution for each slice being between 350 and 400 µm. The 2D images were analyzed using the Avizo^TM^ 3D software from Thermoscientific (Version 2022.1), with an automated Matlab (R2021b release) script followed by a visual check, to identify the different parts of the sample and obtain the 3D morphology of the cracks.

### 2.5. Numerical Modeling

We developed a simple numerical model to complement the function tests. The model was developed using the Abaqus finite element package (2020 release). The model mimics the geometry of the function tests, representing the casing, cement sheath, and sandstone. A plane strain boundary condition is deemed to be most representative for the model (no displacement in the axial direction). The plane strain condition indicates a semi-2D model. Due to the symmetry, only a quarter of the sample is modeled. The model’s geometry is illustrated in Figure 4. The cement is assumed to be fully cured before the pressurization begins. Therefore, cement’s mechanical properties are assumed to have reached their ultimate hydrated values. The model initializes the setup with a small compressive stress. The pore pressure in the system is set to 101.3 kPa (atmospheric pressure). No initial stress at the inner casing or outer surface of the rock is defined. The contacts between the cement and casing/rock are assumed to be fully bonded initially. During the pressurization, a pressure load is applied to the inside of the casing that gradually increases over time. The displacements and stresses in the cement/rock/casing system are recorded over time. If the cement and the rock material fail, the plastic strains are recorded. The mechanical properties of sandstone are measured as part of the experimental campaign and used in the model. Table 2 presents the input parameters used in the modeling. 

The casing elements are assumed to be elastic. The range of stresses in this work makes it unlikely for the steel casing to fail; therefore, casing plasticity is ignored. Triaxial experiments are conducted on the Berea sandstone used in the function tests. These tests provide the elastic mechanical properties, Mohr–Coulomb friction angle, and cohesion data for the rock. The MCC plasticity model was used to capture the failure behavior of cement [30,42]. Equation (1) presents the yield envelope for the MCC model:(1)$$F=\frac{q^{2}}{M^{2}}+\left(p-p_{t}\right)\left(p-p_{c}\right)=0$$
where *q* and *p* are shear and mean effective pressure, respectively; *M* is the slope of the critical state line; *p_t_* is the tensile strength of the sample; and *p_c_* is the initial size of the compressive yield limit. In addition to the yield envelope, Equations (2) and (3) define the normal consolidation and swelling consolidation lines, respectively.
(2)$$\Delta \nu^{p}=\lambda \frac{\Delta p}{p}$$
(3)$$\Delta \nu^{e}=\kappa \frac{\Delta p}{p}$$

In Equation (2), *ν* is the specific volume, *p* is the mean pressure, and *λ* is the slope of the normal consolidation line, which represents the plastic volumetric compression of the material. In Equation (3), *κ* is the slope of the swelling line, representing the volumetric behavior of the sample in the elastic regime. The MCC model (Equation (1)) was implemented as a Fortran subroutine in Abaqus.

The results of the triaxial tests were used to extract the appropriate parameters for the MCC model. The SciPy library in Python was used to fit the best curve to the experimental data in order to estimate the yield envelope parameters in Equation (1). Tensile strength was independently measured for most samples using Brazilian tests. In order to estimate *λ* and *κ*, isotropic compression tests are required. However, such tests were not part of the testing program. Soustelle and coauthors [43] measured the MCC parameters for class G cement, including isotropic compression tests. They reported a value of 0.02 for *λ* and 0.0046 for *κ* for class G cement. *κ* is inversely proportional to the bulk modulus of the material [44]. Therefore, we estimated *κ* for the new formulations tested in this work by the ratio of the bulk modulus between class G cement and the formulations under study. Equations (4) and (5) illustrate the simple scaling scheme used to estimate *κ* and *λ*. *K* is the static bulk modulus of the formulations in this study, measured during triaxial tests.
(4)$$\kappa =\kappa_{classG}\frac{K_{classG}}{K}$$
(5)$$\lambda =\lambda_{classG}\frac{K_{classG}}{K}$$

## 3. Results

### 3.1. Unconfined Mechanical Properties and Sample Water-Fillable Porosity

The mechanical properties of samples exposed in the well for 3 and 9 months are presented in Figure 4, Figure 5 and Figure 6. After the first 3 months of exposure, the uniaxial compressive strength of all tested formulations persisted at an acceptable level of more than 7 MPa, varying between 17 MPa (NAS-M1) and 31 MPa (CAP#71/FAF) (Figure 5, left). The strength loss of about 5% was measured for CAP#71/FAF and CAP#50/FAF. Some initial strength loss followed by strength stabilization has been reported earlier for calcium phosphate-based cements and is not of concern [45]. Surprisingly, the calcium phosphate formulation of #71 cement modified with MK (CAP#71/Silica/MK) increased by 19%. MK is a more reactive pozzolanic component than fly ash F. Its pozzolanic reactions occur even at low temperatures, so if the stabilization of calcium phosphate cement strength involves pozzolanic reactions of FAF, these reactions may happen at earlier curing times with MK, resulting in a strength increase. The formulation of CAC#71 with silica (#71/Silica) lost the most compressive strength (16%). The strength of other formulations increased in the following order: 13% (CSH-60/40) < 25% (NAS-M1) < 50% (TSRC) < 72% (#80/Silica).

The compressive strength of the 9-month exposed samples of the same formulations modified with CMF is shown in Figure 5 (right), along with the strength of the samples before the exposure after 1 day of 300 °C autoclaving. Formulation modification with CMF resulted in a higher initial compressive strength of the samples compared against that of the non-modified formulations exposed for 3 months. This was even though the higher water content was necessary for the same self-leveling of cements with the fibers.

CSH-60/40 experienced a dramatic strength loss of 86% during the 9-month exposure in the well. The strength of the formulation decreased from 35 MPa to 5 MPa. This result was surprising considering the initial strength increase of 13% in CSH-60/40 after the first 3 months of exposure.

As expected for calcium phosphate cements, the strength of CAP formulations with #71 and #50 CAC increased after a longer exposure of 9 months. The strength of CAP#50/FAF more than doubled to the final strength of 35 MPa, while that of CAP#71/FAF increased by 5% to 49 MPa, which was the highest strength among all tested formulations. The strength of CAP#71/Silica/MK dropped by 10%, remaining high at 39 MPa. The increase in strength of the NAS-M1 was for the first 3 months, but the rest of the cement formulations more than doubled in strength: 124% increase for #80/Silica, 130% increase for #71/Silica, and 127% increase for TSRC. The respective final strength values were 33, 45, and 33 MPa. Apart from CSH-60/40, the strength of all the formulations was well above the required minimum of 7 MPa. On average, the strength of the 9-month exposed samples (excluding CSH-60/40) was nearly 40% higher than that of those exposed for 3 months.

One of the major concerns for formulations with very high strength is their brittleness, which can be especially problematic under the conditions of repeated thermal and mechanical stresses typical for HT geothermal wells. YM, as one of the brittleness characteristics [46], can be used to classify cement failure modes from soft (YM < 0.7 GPa) to moderate (YM: 0.7–2 GPa), brittle (YM: 2–3.4 GPa), and very brittle (YM > 3.4 GPa) [16]. For the 3-month exposed samples, according to this classification, samples of CAP#71/FAF and CAP#71/Silica/MK fall into the very brittle failure mode range of more than 3.4 GPa. The NAS-M1 formulation modulus value was in the range of the soft failure mode, while the rest of the formulations were in the range of the moderate failure mode, which is generally desirable under geothermal well conditions, allowing for the avoidance of both brittle and soft cement failures (Figure 6, left).

Often, YM mirrors compressive strength, increasing when the strength increases and decreasing when the strength drops. This was the case for all 3-month exposed samples except for the CAP#71/FAF formulation, which became more brittle despite a 5% decrease in strength. The YM of a similar formulation with CAC#50, on the other hand, decreased by 16% with a 5% strength decrease during the exposure.

The YMs of the 9-month exposed samples were in the brittle range for all samples apart from NAS-M1 with a 0.9 GPa YM, placing it into the moderate failure range, and the CSH-60/40 formulation that experienced dramatic strength and YM loss, falling into the soft mode failure (Figure 6, right). Modification of the samples with CMF resulted in YM values on average about 20% lower than those of unmodified samples, despite 40% higher compressive strength after longer curing under HTHP conditions.

Toughness, which is a combination of strength and ductility, further confirmed this fact (Figure 7). The average toughness of the samples after the 3-month exposure (excluding CSH-60/40) was 0.4 N*mm/mm^3^, while that of the 9-month exposed samples was 1.1 N*mm/mm^3^. Comparison of the reference 300 °C-cured samples to those exposed for 3 months in a well shows that toughness decreased for all the samples except TSR, NAS-M1, and #80/Silica (for the latter, the toughness persisted through the exposure). On the other hand, the toughness of 9-month exposed samples increased after the exposure tests (Figure 7, right). This is likely, at least partially, due to the samples’ modification with CMF. As could be expected, the CSH-60/40 formulation lost most of its toughness, experiencing a 74% toughness decrease.

The water-fillable porosity of the tested formulations is shown in Figure 8. As cement hydration and pozzolanic reactions continue under HTHP conditions, the porosity can be expected to decrease unless samples experience conditions causing their destructive expansion and crack formation. Porosity decreased for all tested formulations both after 3- and 9-month exposures in the well. The average porosity was on the order of 40% for both exposures, with CAP#71/FAF samples showing the lowest porosity (33%) and NAS-M1 showing the highest porosity (51%) after the 9-month exposure. This correlated with the highest and lowest respective compressive strengths of these formulations. Surprisingly, despite the dramatic loss of strength, the CSH-60/40 formulation remained relatively low at 45%.

In general, the mechanical properties of the tested formulations were, for the most part, within the requirements of geothermal well cements. The most commonly used HTHP Portland cement formulation was the only striking exception, losing more than 85% of its strength in 9-month exposure tests.

### 3.2. Mechanical Properties from Tensile and Triaxial Compression Testing

The mechanical properties of the samples exposed in the well for 3 months and the non-exposed samples (control) are presented in Table 3 and Table 4. The triaxial tests confirm the findings of the unconfined compressive strength tests, where both #80/Silica and TSRC samples gained significantly in strength, while CAP#71/FAF shows some slight strength reduction after 3 months of downhole exposure (Figure 9). All 3 formulations become more brittle after exposure, which is manifested in a distinctively more pronounced stress drop when loaded beyond failure and a lesser accumulation of plastic strain at failure. As expected, a clear confining or mean stress dependency of shear stress at failure was observed in all cases and considered in our constitutive model (Section 2.5). #80/Silica and TSRC, and to some degree CAP#71/FAF, show characteristic plastic behavior with strain hardening at higher confining stresses for non-exposed samples. After 3 months of exposure, all materials show strain softening when loaded beyond ultimate stress, and the least so for the TSRC material. When loaded beyond ultimate stress, all samples possess a considerable residual strength, with the magnitude again dependent on the acting mean stress.

During excessive shear loading, the materials show, in most cases, a continuous reduction in pore space because of elastic and inelastic volumetric strain accumulation. Ultimately, many samples develop shear bands with strain localization where the detected volumetric strains suggest a dilatant behavior (Figure 10A). However, this is only the case after significant sample deformation. The higher-porosity TSRC samples exhibit typical compaction bands as a result of pore collapse at high confining pressure (Figure 10B).

The deformation properties of all samples changed after being exposed for 3 months in the Newberry well. YM (E) increased significantly for #80/Silica (+184%) and TSRC (+63%) and, to a lesser extent, also for CAP#71/FAF (+21%), which is qualitatively in line with the results from the unconfined compressive strength tests. From the current dataset, there is no clear trend towards dependency on mean stress. As observed, YM’s values do not change much in either direction.

Poisson’s ratio behavior does not show any clear trends. It is not significantly affected by exposure or mean stress for CAP#71/FAF. The limited data collected for TSRC and #80/Silica are unconclusive. In general, the measured values are all within the expected range for cement materials (0.15–0.25).

The observed property changes can be explained by the alteration of porosity and mineralogical content, as already discussed in Section 3.1.

### 3.3. Function Tests of Selected Cement Formulations

The function cyclic pressure tests were performed on #80/Silica, CAP#71/Silica, and CSH-60/40. Additional information on the function tests of the first two formulations can be found in [40]. The initial scan of the samples showed that they were crack-free. The image analyses of the CHS-60/40 and CAP#71/Silica samples revealed some air-entrained bubbles as small pores occupying about 0.01% of the sample volume. Figure 11 shows the volume of crack appearing in the cement sheath (left) and the rock (right) during the pressure cycles. In both cases, crack formation occurred at the same pressure, which was higher for the #80/Silica formulation than for the other two tested composites (30 and 29 MPa, respectively). After the initiation of the crack propagation, the total crack volume was larger for the #80/Silica formulation. However, the nature of the cracks differed.

CHS-60/40 and CAP#71/Silica showed brittle behavior and underwent instant failure at 29 MPa. #80/Silica, on the other hand, started to crack at 30 MPa, and a full failure was observed at 31 MPa (Figure 12).

The pictures show two different failure behaviors. In CSH-60/40 and CAP#71/Silica, a single crack across the cement sheath and the rock was observed. The crack propagating along the whole sample and through the rock appeared suddenly at 29 MPa. On the other hand, #80/Silica had more ductile behavior, i.e., two small cracks appeared at 30 MPa, and one of them propagated across the rock at 31 MPa.

The crack configurations after the full sample failure are shown in Figure 13. For each test, an example of the 2D horizontal slice is shown, where the different parts of the samples are identified: cement, rock, and cracks. The casing in the center of the picture and the aluminum cell around the rock are not part of this material identification. The uniform red line of the cement sheath failure in the cases of CSH-60/40 and CAP#71/Silica indicates a continuous crack through the cement. In the case of the #80/Silica formulation, cement sheath failure was not continuous (the red color mingled with the white spots), suggesting partial survival of the sheath integrity. This could be important for EGS wells that undergo pressure stimulations. The results show that although the crack volume was smaller in the case of more brittle formulations (one continuous crack instead of several cracks), the propagation of the crack can potentially go further along the casing, creating passages for aggressive fluids and compromising casing corrosion protection by cement. This was not accounted for in the current tests because of the relatively small sample lengths used.

### 3.4. Numerical Results 

#### 3.4.1. Plasticity Parameters

Figure 14 presents the fitted curve to the triaxial and UCS data for the CAP#71/FAF sample. If the loading test (triaxial or UCS) indicates strain softening, i.e., a drop in stress after the peak is reached, then the peak indicates a point on the yield envelope. The residual stress after the peak indicates a point on the critical state line. Strain softening is indicative of the “dry side” of the yield envelope on the left of the critical state line, as is seen in Figure 13. If strain hardening behavior is observed in the experiments, then the yield envelope is reached at the onset of deviation from linear elasticity. Residual stress is reached as axial stress reaches a constant value, indicating the point on the critical state line. Table 4 presents the estimated MCC parameters for the Newberry well and control samples.

The tensile strength values in Table 4 are inferred from Brazilian tests on the Newberry well and control samples for all the formulations except the CAP#71/FAF samples. The TSRC (control) sample shows the closest MCC parameter values to the class G cement, as reported by [43]. The TSRC sample exposed to the Newberry well condition shows an expanded yield surface and an increase in the slope of the critical state line (M). Tensile strength also shows a modest increase. This indicates an increase in the overall strength and stiffness of the sample due to the exposure. The CAP#71/FAF samples show relatively constant MCC parameters before and after exposure to the well conditions. The size of the yield envelope is significantly larger for the CAP#71/FAF sample compared to class G cement (approximately 100 MPa compared to 23 MPa). The #80/Silica (control) sample indicates a slight increase in strength compared to class G cement; however, exposure to well conditions marks a significant increase in strength and stiffness. The size of the yield envelope increases from 28 to 58 MPa, while M increases from 1.3 to 1.6. The tensile strength increases significantly from 3.45 to 5.13 MPa. These results indicate that the present cement formulations may show different mechanical behavior depending on their age and temperature conditions. Well integrity models should take this into account in order to improve modeling predictions. If the short-term behavior of the cement is of interest, the mechanical parameters of the control samples are likely more representative. However, if the long-term behavior of cement is being modeled, the Newberry sample results should be used in numerical models.

#### 3.4.2. Function Test Model

The results of the numerical model replicating the function tests are presented in this section. The model is run for six formulations as presented in Table 4, along with class G cement with MCC parameters reported by [43]. The numerical model tracks the evolution of plastic strains in the cement elements as the casing is pressurized. Figure 15 demonstrates the average value of the volumetric plastic strain versus casing pressure. The general trend of the results shows no initial plastic strain in the cement. At a certain casing pressure, plastic strains start accumulating. This is equivalent to the onset of cracking in the function tests. The positive value indicates dilation or an increase in the sample volume, indicative of cracking. The numerical results also show failure in the sandstone, which is noted in the experimental observations as well.

The reference class G cement shows initial signs of cracking at a casing pressure of 18 MPa. The #80/Silica (control) sample shows initial signs of failure at 31 MPa. This is similar to the experimental observation. The #80/Silica (Newberry) sample starts cracking at the same casing pressure of 31 MPa. This is despite the fact that exposure to high-temperature conditions and aging increases the strength of this formulation. The increase in strength comes with an increase in stiffness, which can be detrimental as it leads to higher levels of stress in the cement. In this case, the increase in stiffness compensates for the improvement in strength. The final results show more brittle cracking in the exposed sample, indicated by a slightly higher level of plastic volumetric strain.

The CAP#71/FAF (control) has the largest yield envelope of the three formulations in Table 4. However, it also has the highest Young’s modulus. Therefore, the casing pressure increase can lead to high stresses in the cement sheath. CAP#71/FAF (control) shows initial signs of cracking at 31 MPa. In contrast, CAP#71/FAF (Newberry) starts cracking at 23 MPa. The reason for the earlier failure of CAP#71/FAF (Newberry) is the higher stiffness compared to the control sample (19 versus 16 GPa). Once CAP#71/FAF (Newberry) fails, the rate of increase in plastic strain is the highest of all samples, indicating a high level of brittleness.

The TSRC (control) formulation cracks at a pressure of 29 MPa. TSRC (Newberry) shows similar behavior and starts cracking at 29 MPa casing pressure. However, the rate of plastic strain accumulation is higher for the Newberry sample, similar to the #80/Silica formulation. Overall, all the formulations considered here performed better than class G cement, accumulated less plastic strain, and initiated cracking at a higher casing pressure. Exposure to the Newberry well conditions improved the strength of the formulations. However, it also increased the stiffness (Young’s modulus) of the formulations, which may lead to more brittle behavior and therefore higher levels of crack volume after failure.

### 3.5. Phase Compositions and Morphologies of Selected Formulations

To understand changes in the mechanical properties of the exposed formulations and to predict their further stability and degradation under the well conditions, phase identification and morphological studies were performed for the designs of interest. Changes in the Portland cement-based formulation, CSH-60/40, that caused dramatic loss of mechanical properties without increasing the sample’s porosity were of considerable interest.

#### 3.5.1. CSH-60/40

Figure 16 shows the XRD patterns of the reference CSH-60/40 sample and those exposed in the well for 3 or 9 months.

For the CSH-60/40 formulation, the main crystalline phases of hydrated cement after one day of autoclaving at 300 °C were predictably tobermorite and xonotlite. Xonotlite should be the dominant crystalline phase at this temperature, with tobermorite temperature stability being below 200 °C [47]. Nevertheless, tobermorite was clearly present in the reference sample, although the intensity of xonotlite peaks was higher than for tobermorite ones. Further conversion of tobermorite to xonotlite after longer HT exposure was expected. This conversion is accompanied by microstructural changes with the growth of xonotlite needle crystals, resulting in slightly decreased strength and increased porosity. However, mechanical property analyses of this cement after the 3-month exposure revealed decreased sample porosity and increased strength. This unusual behavior can be understood from the results of the composition analyses.

For the field samples, xonotlite peaks were nearly absent from the XRD patterns. The patterns were dominated by the peaks of calcium carbonate and non-reacted silica. For the most part, the crystalline calcium–silicate phases converted to calcium carbonate. Only small xonotlite shoulders were still visible in the pattern of the 3-month exposed sample (e.g., 2 Ɵ ~28.8), but they completely disappeared from the patterns of the 9-month exposed samples. The only other crystalline phase detected by XRD was silica. The intensity of silica peaks decreased, suggesting that it, at least partially, participated in the tobermorite-to-xonotlite conversion before the sample was carbonated. The initial cement carbonation resulted in increased sample strength and decreased porosity due to the matrix densification with calcium carbonate.

However, after the longer 9-month exposure, calcium removal from calcium–silicate hydrates through carbonation resulted in a dramatic strength decrease. In the excess carbon dioxide and water vapors, calcium carbonate is converted into soluble calcium bicarbonate, which can migrate out of the cement matrix, leaving amorphous silica gel behind.

Strong sample carbonation was confirmed by the TGA/DTG tests (Figure 17). The only large decomposition event for both the 3- and 9-month exposed samples is carbonate’s decomposition. The mass loss corresponding to this step is 21% and 28% after the 3- and 9-month exposures, respectively. A small decline in the weight curve associated with the cement hydrates was still visible after 3 months in the well, while no weight loss associated with the cement hydrates was detected after the 9-month exposure. Assuming the decarbonation weight loss was only due to the calcium carbonate decomposition and knowing the initial weight percent of CaO in the class G cement (73%), the mass loss of 21% for the 60/40 cement/silica formulation means that 54% of the original calcium in the class G cement was in the form of carbonate after the 3 months of exposure. The 28% CO_2_ loss during the decarbonation step of the OPC/silica formulation with 5% CMF corresponds to 83% of the calcium in the original formulation being carbonated. Considering that calcium carbonate was likely partially converted into the soluble calcium bicarbonate that could migrate from the samples, even higher conversion of the original calcium in the exposed samples is likely.

Figure 18 shows the morphologies of the 3- and 9-month exposed samples, and Table 5 provides the results of elemental composition measurements in representative locations.

The SEM images of the 3-month exposed sample show large calcium carbonate crystals (location 1) formed in an otherwise mostly amorphous cement matrix and smaller calcium carbonate crystals of different sizes (location 2) embedded into the matrix throughout the sample. In agreement with the XRD data, xonotlite needle-like crystals were not detected. Although well-formed, the large calcium carbonate crystals did not compromise the mechanical properties of the 3-month exposed samples. However, their precipitation did not prevent further sample carbonation, blocking CO_2_ and water penetration into the sample. The large crystals in the images of the 9-month exposed samples belonged to non-reacted crystalline silica (location 4). Calcium carbonate crystals were not observed, the morphological features of the matrix became smaller, and the structure of the matrix had a porous aspect that would be favorable for continuous carbonation of the sample. The elemental composition in other tested locations corresponded to the calcium bicarbonate present along with silica gel (locations 3 and 5).

In summary, for the CSH-60/40 formulation, the results of all the analyses agreed. This formulation underwent severe carbonation in the well, resulting in its loss of mechanical properties. Its water-fillable porosity nevertheless persisted, with calcium carbonates forming a matrix with relatively low permeability.

#### 3.5.2. Calcium Phosphate Cement with Different CAC Grades (CAP#71/FAF, CAP#50/FAF)

Calcium phosphate cement formulations were specifically developed to withstand CO_2_-reach HT environments in geothermal wells [13,48]. The CO_2_-resistance of this cement comes from CO_2_ mineralization with the formation of stable carbonated apatite and cancrinite phases [49]. The two formulations of CAP cement with FAF were made with CAC#71 and with CAC#50. CAC#71 has a higher aluminum content (55.8 wt.% Al_2_O_3_) and a lower Ca content (44.0 wt.% CaO) than CAC#50 (45.1% and 49.7%, respectively). Figure 19 and Figure 20 show XRD patterns of these formulations for the reference samples and samples exposed in the well for 3 or 9 months.

The reference patterns include expected phases of boehmite (aluminum oxide hydroxide), hydroxylapatite and analcime, high-temperature stable zeolite, and the feldspar mineral dmisteinbergite (CaAl_2_Si_2_O_8_), an isomorph of anorthite. In the formulation with CAC#50, which is richer in calcium, crystallization of katoite takes place (Ca_3_Al_3.5_O_4.5_(OH)_7.5_). This phase is absent in CAP#71/FAF with more aluminum-rich CAC#71.

Samples exposed to the well resulted in the disappearance of analcime peaks in both formulations and the disappearance of katoite peaks in CAP#50/FAF. The intensity of boehmite peaks strongly dropped in CAP#50/FAF with lower aluminum content but remained strong in CAP#71/FAF. The intensity of the dmisteinbergite peaks decreased in both formulations. The new peaks of mica-type minerals appeared in the patterns. These were identified as belonging to paragonite, margarite, muscovite, and Ca-mica minerals. Calcium carbonate peaks were present in the patterns of both formulations.

The peaks of hydroxylapatite were clearly present in the patterns of both formulations. However, the expected carbonated phases of cancrinite and carbonated apatite were not found. If for carbonated apatite, phase identification is somewhat problematic due to the strong patterns overlapping with the apatite phase peaks and the low crystallinity of the newly formed carbonated apatite phase during CAP cement carbonation, the phase of cancrinite could be identified if it formed in the samples.

Partial carbonation of the samples was confirmed with TGA/DTG and EDX analyses (Figure 21). The two major weight loss events in thermogravimetric experiments of CAP cement formulations were the decomposition of boehmite between 400 and 550 °C and the decarbonation of the samples above 600 °C [50]. The weight loss associated with boehmite dehydroxylation was 5% and 4% after 3 and 9 months of well exposure for CAP#71/FAF and 3 and 1.5% for CAP#50/FAF, respectively (Figure 22). This result agreed with the strongly decreased peak intensities of boehmite in 9-month exposed CAP#50/FAF samples. The extent of carbonation for CAP#71/FAF formulation did not change between 3 and 9-month exposures, persisting at 8%.

For CAP#50/FAF, the decarbonation peak was slightly smaller after 9 months in the well (9% vs. 11% after the 3-month exposure). The decrease in carbonate concentration during longer exposures could be attributed to the continued carbonation and removal of soluble carbonates from the sample. However, it is likely that samples did not undergo any significant additional carbonation, as in the case of the CSH-60/40 formulation, since the percent of decarbonation did not increase.

The results of morphological analyses are shown in Figure 22 and Figure 23, and the EDX compositions along with the phases identified in selected locations are shown in Table 6 and Table 7.

In agreement with the XRD and TGA/DTG data, the CAP#71/FAF formulation had some inclusions of calcium carbonate crystals after the 3-month exposure (location 1). The matrix also still contained non-reacted FAF particles (top right photomicrograph, Figure 23) and clearly identifiable dmisteinbergite crystals (location 2). After the 9-month exposure, the larger crystalline features similar to dmisteinbergite and more compact embedded into the dense matrix had compositions related to mica-type minerals, margarite, with its typical morphology of a mass with thin laminae, and Ca-mica (locations 3 and 4).

The photomicrographs of CAP#50/FAF samples are presented in Figure 24, and Table 7 shows EDX elemental analyses at specified locations and possible corresponding phases. The 3-month well-exposed sample matrix was very dense and rich in aluminum and silica, with the presence of phosphorus, calcium, and some iron (location 1). Carbon detection indicated partial matrix carbonation; however, large calcium carbonate crystals were not found in the sample. Non-reacted particles of FAF were still visible (photograph, top right) embedded into the matrix. EDX analyses showed that phosphorus phases made up part of small crystals of less than 1 micron or were incorporated into the amorphous matrix. After the 9-month exposure, a dense matrix was formed with tiny crystals of calcium mica (locations 2, 3 and 4). The photomicrograph also shows intact carbon fibers that withstood a 9-month exposure in the bulk cement without any visible damage.

In summary, CAP cement formulations underwent only partial carbonation during the well exposure tests. The crystalline composition of these samples persisted through the exposure, with the major crystalline phases of dmisteinbergite, paragonite, and hydroxylapatite remaining in the samples after the 9-month exposure. The extent of the carbonation was higher for the more Ca-rich CAP#50/FAF formulation than for the CAP#71/FAF one with a lower calcium content. Persistent crystalline compositions and a dense matrix with limited carbonation provided improved mechanical properties and a very low water-fillable porosity for these formulations.

#### 3.5.3. TSRC

Like CAP cement, TSRC was developed to withstand high thermal shocks typical for HT geothermal wells and was also expected to mineralize CO_2_ into a stable cancrinite phase [49]. This blend is based on CAC#80 with the lowest calcium (24.7 wt.%) and highest aluminum (75.2 wt.%) contents. This Al-rich composition, combined with the FAF, provides high thermal shock resistance for the blend.

Figure 25 presents the XRD patterns of the TSRC reference samples and samples exposed in the well for 3 or 9 months. The reference sample crystalline composition was very similar to that of CAP/FAF formulations, except for the phosphorus-containing phases that were absent in TSRC and the presence of corundum (aluminum oxide) crystals from CAC#80 in the TSRC formulation. The major crystalline phases include dmisteinbergite and its isomorph anorthite, analcime, katoite, boehmite, and non-reacted mullite from FAF. After the 3-month well exposure, analcime, katoite, and mullite peaks disappeared, while new peaks of paragonite, margarite, and calcium carbonate showed up in the pattern, and peaks of boehmite persisted. After the 9-month exposure, boehmite peaks vanished from the sample pattern, while the peaks of paragonite, margarite, and calcium carbonate persisted. Although the patterns of dmisteinbergite overlap with other phases identified in the exposed samples, its peaks at 2θ of 24.08 and 31.5 were clearly identifiable in the exposed samples. The persistence of crystalline phases implies their stability over the exposure time under the well conditions, which is further supported by the similarity of the patterns of the 3- and 9-month exposed samples.

The TGA/DTG analyses confirmed partial carbonation of the sample (Figure 26). In agreement with the XRD data, the two main weight loss events were from the dehydroxylation of boehmite (1% weight loss in the 3-month exposed sample and 0.4% weight loss in the 9-month exposed sample) and decarbonation of the samples (8 and 7%, respectively). These data confirmed the disappearance of boehmite after longer well exposure and the persistence of carbonation, which, unlike for the CSH-60/40 formulation, did not increase after longer exposure. The slight decrease in decarbonation weight loss could be attributed to the partial dissolution of the carbonates.

Figure 27 shows the microstructures of the exposed samples, and Table 8 provides their elemental compositions in selected locations along with the suggested phases. The morphological study of the 3-month exposed samples showed non-reacted FAF particles (top left photomicrograph) remaining in the samples. Pozzolanic reactions of these particles provide self-healing properties of the cement, so their presence in the cement matrix suggests that cement still possessed self-healing properties after the 3-month exposure. In agreement with the XRD data, these samples contained boehmite plates (location 1) and margarite crystals embedded into the dense matrix (location 2). The 9-month exposed samples showed feldspar minerals anorthite (location 3) and dmisteinbergite (location 4). The cubic crystal surrounded by dmisteinbergite plates is calcium carbonate. The possible pathway for such close co-existence of these crystalline structures is the transformation of dmisteinbergite into calcium carbonate during the longer exposure. As in the CAP#50/FAF samples, there was no visible damage to carbon fibers from the exposure of TSRC, despite their 2–3 orders of magnitude higher pore water pH than that of CAP#50/FAF.

In summary, the TSRC formulation, designed to withstand the HT thermal shocks of geothermal wells, was also stable under the conditions of the Newberry well. Its crystalline composition persisted, and carbonation was limited, which resulted in improved mechanical properties and decreased water-fillable porosity. Moreover, the persistence of FAF in the composition of the blend suggests that it kept its self-healing properties for months at very high temperatures.

#### 3.5.4. #80/Silica

This simple two-component blend showed significant improvement in its mechanical properties (especially toughness) and decreased water-fillable porosity during the exposure tests. Refractive CAC#80 is well-suited for the HT geothermal conditions [17]. The XRD patterns of the reference sample (1 day at 300 °C) and samples exposed to 3- and 9-month well conditions are shown in Figure 28.

The pattern of the reference sample included CAC#80 hydration product boehmite, CAC#80 and silica reaction product anorthite, and non-reacted phases corundum from CAC#80 and silica. Exposure of the sample to the Newberry well conditions resulted in the formation of anorthite isomorph dmisteinbergite after the 3-month exposure and calcium silicate hydrate (Ca_6.43_(Si_2_O_7_)_2_(H_2_O)_2_), as well as unnamed zeolite (K_2.84_Ca_1.43_Al_5.7_Si_10.3_O_32_*10.6H_2_O) after the 9-month exposure. Since the intensity of anorthite peaks was similar after 3 and 9 months, while that of dmisteinbergite decreased after the 9-month exposure, it is reasonable to think that the dmisteinbergite isomorph underwent partial conversion into albite while anorthite persisted. In the unnamed zeolite cation, calcium is replaced by potassium. The original formulations did not contain any alkaline activators, which means that potassium ions came from the well fluids. A small peak of calcium carbonate appeared in the field exposed samples. The TGA/DTG tests showed that decarbonation accounted for less than 2% of mass loss after 3 months of the well exposure and less than 1.5% mass loss after 9 months of the exposure [19].

The morphological study of the samples confirmed the XRD results (Figure 29, Table 9). Dmisteinbergite (location 1) and boehmite (location 2) were detected in the 3-month exposed sample. Partial degradation of the boehmite crystals with the formation of a fluffy amorphous phase around them is visible in the photomicrograph (location 2). The 9-month exposed samples showed sites with typical anorthite morphology and elemental composition (location 3) and the elemental composition of the unnamed zeolite around the crystals with dmisteinbergite morphology (location 4). Boehmite crystals were not detected in the 9-month exposed samples, in agreement with the XRD results. The morphological features of these samples were small. As for other formulations, intact carbon fibers were visible in the 9-month exposed sample.

In summary, the #80/Silica formulation showed good resistance to the conditions of the HT geothermal well. The feldspar minerals formed during the blend hydration were partially converted into the alkali plagioclase series member albite and an unnamed zeolite, with the alkaline ions coming from the well environment. The carbonation of the blend was minimal.

## 4. Discussion

Exposure of various cementitious blends in a deep HT geothermal well for up to 9 months allowed evaluation of their performance under field conditions that would be very difficult to reproduce in laboratory tests. Although the samples were originally cured hydrothermally in laboratory environments (1 day at 300 °C), the general tendencies in the behavior of different blends can be deduced from the results of the exposure tests. Among the tested formulations, only CSH-60/40 was a calcium–silicate blend of OPC and silica. The rest included CAC, and NAS-M1 was calcium-free. The well environment was rich in CO_2_, with a well temperature and pressure (300–350 °C and 26 MPa, respectively) indicative of its supercritical state. It is likely that other geological fluids/gases were also present in the well.

Even though all tested blends underwent phase transitions under the well conditions over the long exposure times, all of them except the CSH-60/40 maintained, and often improved, their mechanical properties and decreased water-fillable porosity after the 9-month exposure. Although persisting, the mechanical properties of the NAS-M1 sample remained lower than for other Al-rich formulations, while the porosity was higher. Optimization of this formulation will require further efforts.

The dramatic loss of CHS-60/40’s mechanical properties (86% strength loss) after the 9-month exposure coincided with the blend’s severe carbonation (83% of the original calcium was carbonated based on the results of TGFA analyses). If the initial blend carbonation with precipitation of calcium carbonate in the pores after the 3-month exposure resulted in improved strength (13% increase) and decreased porosity (24% decrease), further carbonation compromised the samples’ performance through the formation of soluble calcium bicarbonate and amorphous silica gel according to the following reactions:CO_2_ + H_2_O = H^+^ + HCO_3_^−^
C-S-H phase + H^+^ + HCO_3_^−^ = CaCO_3_ + amorphous gel
CaCO_3_ + CO_2_ + H_2_O = Ca(HCO_3_)_2_

OPC/silica decomposition due to carbonation in less than 9 months significantly disagrees with multiple other works cited in the introduction on Portland cement carbonation rates. This is likely due to the severity of the well environment, where HT supercritical CO_2_ reactions with the cement were dramatically accelerated. It should also be mentioned that the samples were relatively small in volume (12.6 cm^3^), facilitating their carbonation. Nevertheless, the long-term stability of OPC-based cements under such conditions could hardly be expected, even for larger cement volumes.

Formulations based on CAC all experienced partial carbonation through the calcium reaction, resulting in the formation of calcium carbonate. These formulations did not form any calcium–silicate hydrates after the initial 1-day curing at 300 °C. They all formed the plagioclase minerals anorthite and dmisteinbergite. The HT carbonation of anorthite/dmisteinbergite can be summarized in the following reaction [51,52]:CaAl_2_Si_2_O_8_ + CO_2_ + 2H_2_O = CaCO_3_ + Al_2_Si_2_O_5_(OH)

The reaction proceeds in solution after the dissolution of carbon dioxide and anorthite [53].
CO_2_ + H_2_O = H^+^ + HCO_3_^−^
CaAl_2_Si_2_O_8_ + 16H^+^ = Ca_aq_^2+^ + 2Al_aq_^3+^ + 2SiO_2aq_ + 8H_2_O

Boehmite precipitates first after the anorthite/dmisteinbergite dissolution due to its very low solubility.
Al_aq_^3+^ + 2H_2_O = AlO(OH)_s_ + 3H^+^

The precipitated boehmite may be very fine-grained and porous; it may also contain silica in its composition as contamination from anorthite [53,54]. Such modified boehmite covers plagioclase grains, slowing their further dissolution. The next phase that precipitates out of solution is calcium carbonate, which has a higher solubility than boehmite. Finally, kaolinite precipitates in the reaction between aluminum and silica.

However, in the case of the well-exposed samples, formulations included sodium-based activators. This allowed for the formation of mica-type minerals (paragonite, margarite, and muscovite) rather than kaolinite. Moreover, alkaline ions were likely found in the well environments, since even the formulation of #80/Silica that did not have any activator had potassium in its composition (in the unnamed zeolite).

Anorthite/dmisteinbergite phases persisted in all the tested CAC-based formulations, possibly with the partial carbonation causing fine-grained modified boehmite precipitation around these crystals. This nanoscale, fine-grained boehmite was likely not detected by the XRD measurements. Such an amorphous phase was visible in the #80/Silica sample around partially decomposed boehmite crystals after the 3-month exposure. The originally formed boehmite, on the other hand, disappeared in all the formulations except CAP#71/FAF. It is not clear why one of the CAC-based formulations preserved crystalline boehmite after the 9-month exposure but not the others.

Based on the TGA analysis, the amount of calcium carbonate formed in each blend depended on its calcium content. For the CAC-based formulations, it decreased in the following order: CAP#50/FAF > CAP#71/FAF = TSRC > #80/Silica. The calcium carbonate content of these blends slightly decreased over time (9-month vs. 3-month data). This was likely due to the dissolution of some of the calcium carbonate through continuous carbonation and the formation of calcium bicarbonate. The decrease in calcium carbonate was 18% for CAP#50/FAF, 15% for #80/Silica, and 12% for TSRC. CAP#71/FAF decarbonation weight loss did not change. Most importantly, unlike in the case of CSH-60/40, CAC-containing formulations still preserved stable crystalline phases, such as mica-type and plagioclase minerals, after the partial carbonation of the matrix. These phases included minerals from the mica family and the end member of the plagioclase series, albite. Moreover, carbonation of calcium–plagioclase minerals (anorthite and dmisteinbergite) was only partial over the experimental period. These allowed for the preservation of the mechanical and physical properties of the CAC-based formulations. Carbon fibers tested as part of cement compositions in the well for 9 months preserved their physical integrity, contributing to the samples’ strength and toughness.

The performance of the function cyclic pressure tests allowed for the comparison of cement formulations in laboratory experiments. Although intended for high temperatures, CAC-based formulations tested in function tests after the 110 °C curing were observed to be comparable to or outperforming the OPC/silica blend cured under the same condition. The failure mechanism of the cement sheath and rock fracture depended on the brittleness of the cement formulation, with more brittle formulations failing at once with a single crack and more ductile formulations failing over a range of pressure, developing multiple fractures.

Although the function tests do not directly apply to the field conditions, the corresponding modeling results confirm that the MCC model and the measured parameters can likely be extended to the field conditions. Staged finite element models can be used to estimate the performance of such formulations under realistic conditions [55]. The benefit of the MCC parameter is that it can predict both the dilatative and compressive volumetric responses of the cement as it fails. Dilatative failure, as shown in the function tests, likely leads to permeability generation. However, compressive volumetric response causes a reduction in the volume of cement and no crack development. Therefore, it is important to distinguish the post-failure response of the cement sheath to assess the consequences in terms of well leakage. The MCC parameters reported in this work provide the necessary values to conduct such an assessment for a variety of cement formulations.

The modeling results of the function tests indicate an overall increase in the brittleness of the Newberry-exposed cement formulations. Long-term exposure to critical temperatures increases strength and stiffness (Young’s modulus). The increase in stiffness leads to higher stress concentrations. This causes relatively earlier damage initiation and more crack volume development under the conditions of the function tests. Therefore, it is plausible that the current formulations can gradually become more brittle and prone to cracking under in-situ conditions in high-temperature geothermal wells.

It should be noted that all material analyses were conducted at room temperature after exposure to HTHP conditions. In future work, in-situ material characterization and measurements of their mechanical properties under well conditions would be beneficial.

## 5. Conclusions

The performance of various cementitious composites under the conditions relevant for geothermal wells was evaluated in function cyclic pressure tests and HT geothermal well exposure tests and compared against that of the OPC/silica blend. The tested formulations included calcium–aluminate cement (different grades) blends with silica or fly ash F. Some of the blends included alkali (SMS in TSRC) or chemical activators (SHMP in CAP cement blends). Additionally, blends reinforced with CMF were tested in 9-month field exposure tests. The paper reports the results of the unique exposure tests of various cementitious composites in a deep geothermal well, allowing for a correlation between the laboratory tests and the actual field performance of the tested composites.

The results of all the tests and cement modeling at temperatures ranging from 120 °C to above 300 °C showed that CAC-based blends outperform the reference OPC/silica one. Interestingly, the findings of the short-term (3-month) and long-term (9-month) exposures to the Newberry well conditions differed in an important way. The short-term exposure increased the shear and tensile strength of most of the tested formulations, including the control OPC/silica one. Calcium phosphate cement formulations, however, experienced a slight reduction in shear and no change in tensile strength. This loss was recovered after the 9-month exposure. On the other hand, the 9-month exposure led to a very substantial loss of strength, stiffness, and toughness for the OPC/silica blend. This happened because of the fast degradation of calcium–silicate hydrates through carbonation, resulting in compromised mechanical properties. The extent of partial carbonation in the CAC-based blends depended on their calcium content. The carbonation in these blends took place through the removal of calcium from plagioclase end-series member anorthite (dmisteinbergite) and the formation of the end-family member albite and mica family minerals margarite, muscovite, and paragonite. These mineral phases allowed for the persistence or improvement of the mechanical properties of the samples during their well exposure. Carbon microfibers persisted in cementitious composites through the 9-month exposure without any visible degradation, improving their strength and toughness.

The Modified Cam-Clay plasticity parameters of several HT cement formulations were extracted from triaxial and Brazilian tests and verified against the experimental results of function cyclic tests. These parameters can be used in well integrity models to predict the field-scale behavior of the cement sheath under geothermal well conditions. Overall, all the considered formulations performed better than OPC, accumulating less plastic strain and initiating cracking at a higher casing pressure. Exposure to the Newberry well conditions improved the strength of the formulations. However, it also increased stiffness for all formulations after the three months of exposure, which may lead to more brittle behavior and therefore higher levels of crack volume after the failure.

The results clearly demonstrate that currently used OPC/silica formulations are not stable under the conditions of HT geothermal wells, where the presence of geological gases is likely or can be expected after well stimulations in the case of EGS. The results of low-temperature laboratory studies on OPC/silica formulation stability under high CO_2_ concentrations are not applicable to HT geothermal conditions. Moreover, function cyclic stress tests at lower temperatures and modeling results showed that this formulation is outperformed by calcium–aluminate-based cement designs over the whole range of test conditions and environments reported in this paper.

## Figures and Tables

**Figure 1 materials-17-01320-f001:**
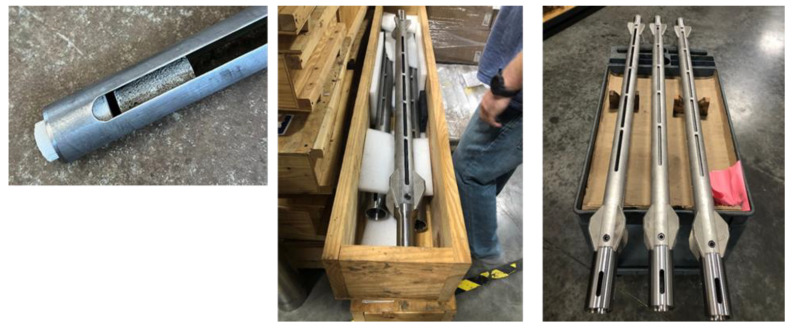
Photographs of sample exposure tools (baskets) with cylindrical cement samples inside.

**Figure 2 materials-17-01320-f002:**
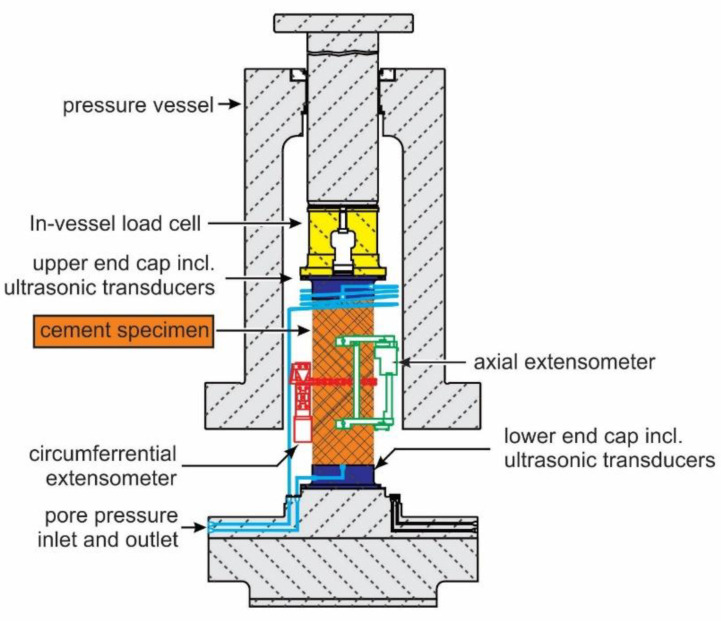
A setup for triaxial testing of cement specimens.

**Figure 3 materials-17-01320-f003:**
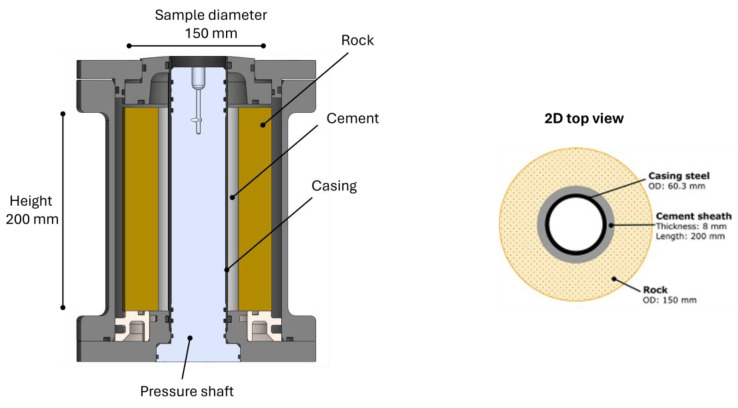
Schematic illustration of a small-scale wellbore cell used to perform function tests (**left**) and a 2D illustration of a sample with a cement sheath in the annuli between rock and casing (**right**).

**Figure 4 materials-17-01320-f004:**
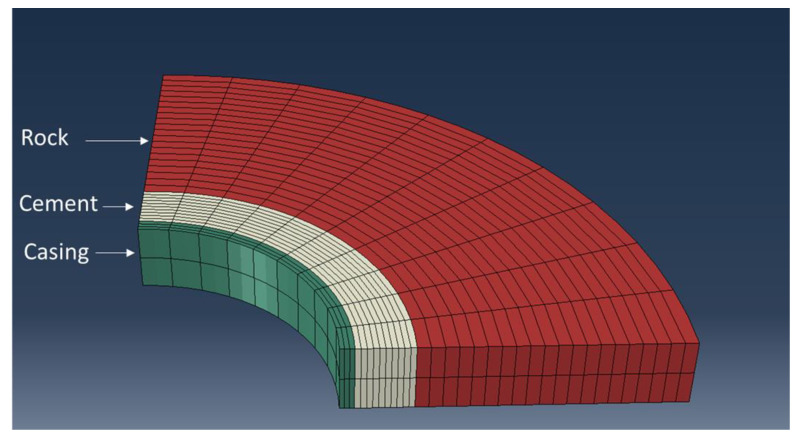
Geometry of the numerical model, representing a semi-2D slice of the sample used in the function tests.

**Figure 5 materials-17-01320-f005:**
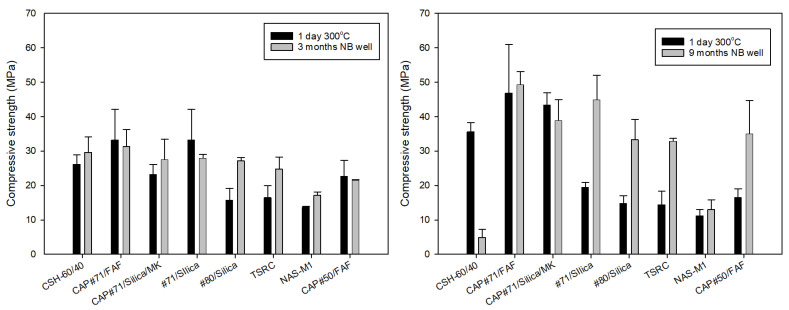
Compressive strength of reference samples (after 1 day of autoclaving at 300 °C) and samples exposed in the Newberry well for 3 months (**left**) or 9 months (**right**).

**Figure 6 materials-17-01320-f006:**
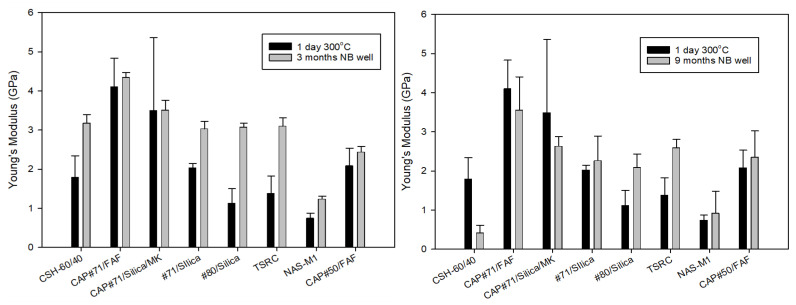
Young’s modulus of reference samples (after 1 day of autoclaving at 300 °C) and samples exposed in the Newberry well for 3 months (**left**) or 9 months (**right**).

**Figure 7 materials-17-01320-f007:**
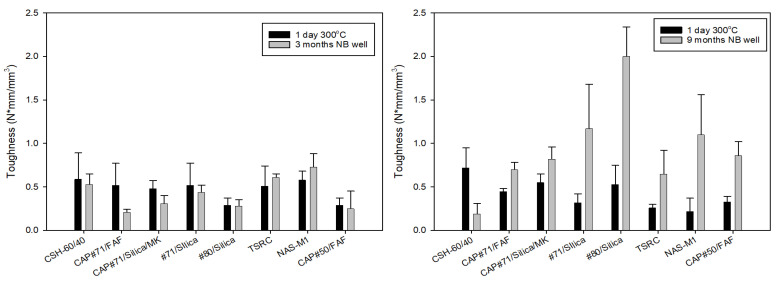
Toughness of reference samples (after 1 day of autoclaving at 300 °C) and samples exposed in the Newberry well for 3 months (**left**) or 9 months (**right**).

**Figure 8 materials-17-01320-f008:**
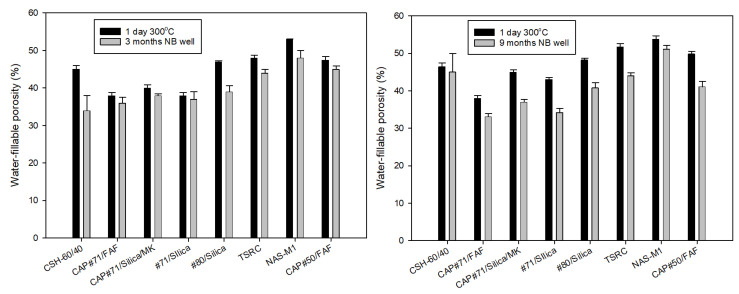
Water-fillable porosity of reference samples (after 1 day of autoclaving at 300 °C) and samples exposed in the Newberry well for 3 months (**left**) or 9 months (**right**).

**Figure 9 materials-17-01320-f009:**
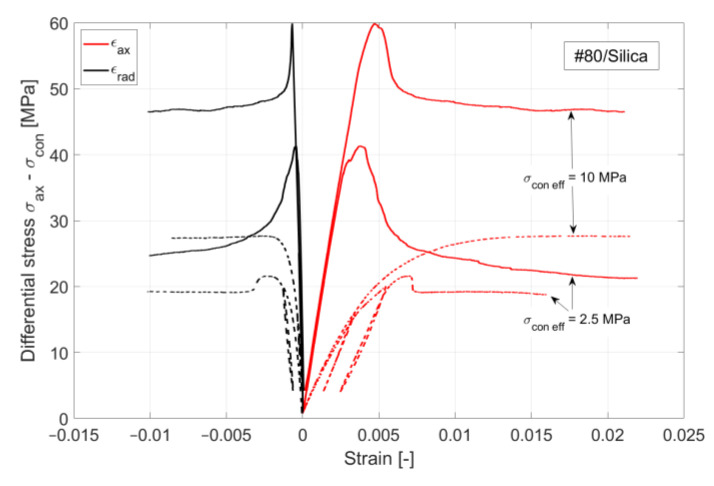
Stress–strain plots from triaxial testing for three different materials. Solid lines correspond to exposed samples (3 months, Newberry well), and dashed lines are the non-exposed control samples (1 day of autoclaving at 300 °C).

**Figure 10 materials-17-01320-f010:**
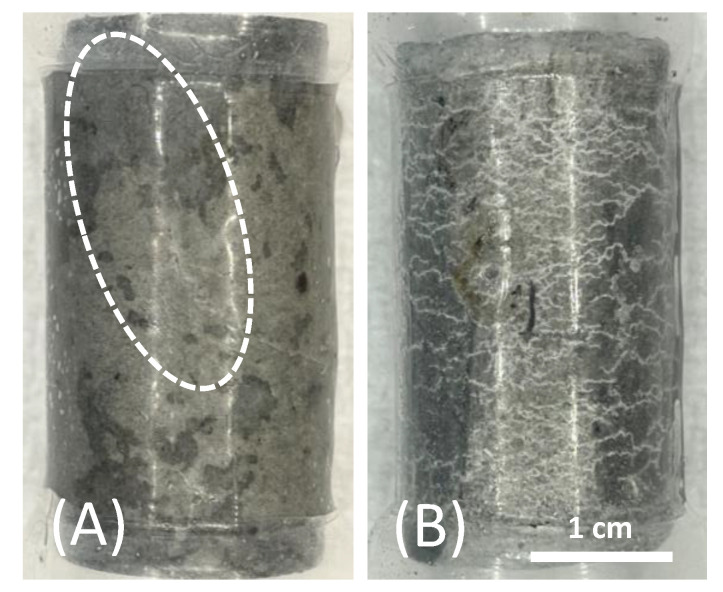
(**A**) Formation of shear bands associated with volumetric strain expansion during triaxial testing for a CAP#71/FAF sample (3 months exposure, Newberry well) at 2.5 MPa effective confinement. The trace of a steeply inclined shear band can be seen in the middle of the stippled white ellipse. (**B**) Typical signs of pore collapse (formation of numerous compaction bands) for a TSRC sample (1 day of autoclaving at 300 °C) tested at 10 MPa effective confinement.

**Figure 11 materials-17-01320-f011:**
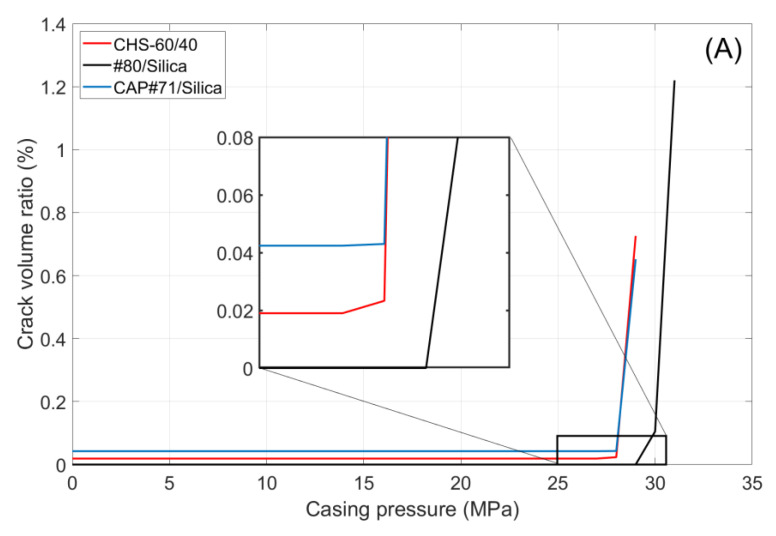
Crack growth in cement and rock during the function tests for (**A**) the cement sheath and (**B**) the surrounding rock.

**Figure 12 materials-17-01320-f012:**
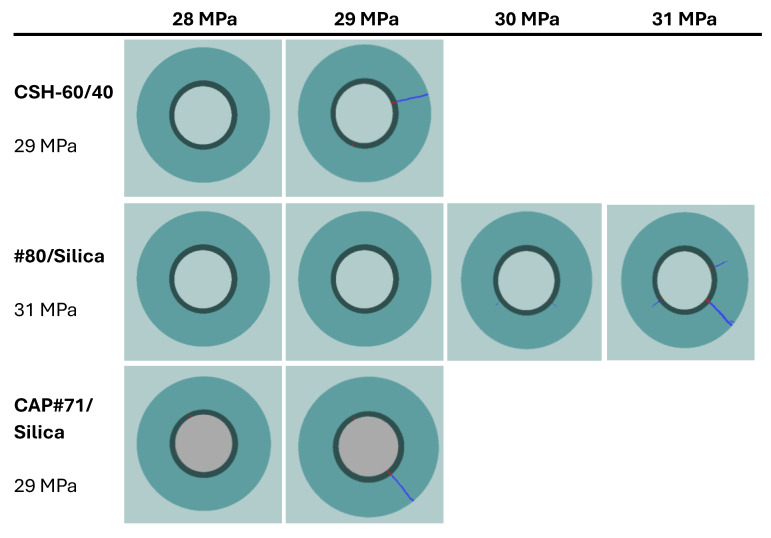
2D segmented images illustrating the brittle behavior of CAP#71/Silica and CSH-60/40. Material identification on the 2D picture: cement (dark green), rock (light green), cement crack (red), and rock crack (blue).

**Figure 13 materials-17-01320-f013:**
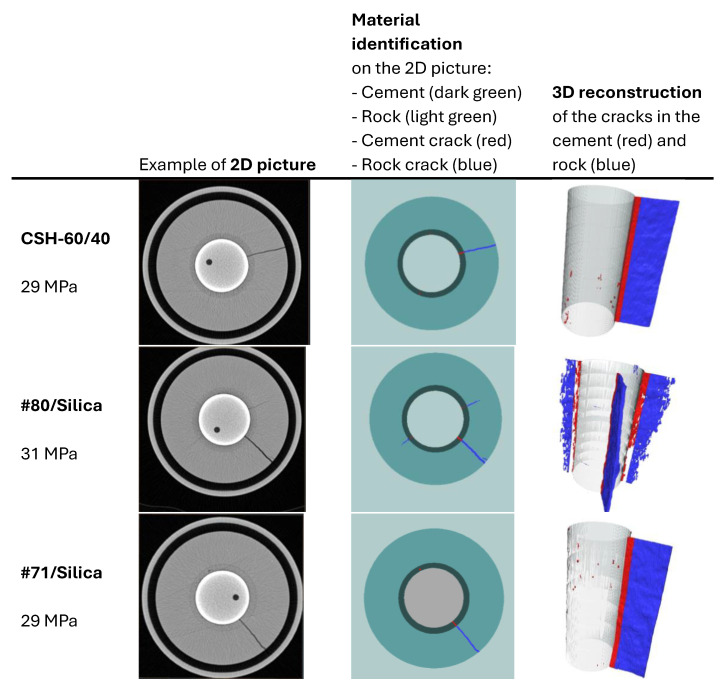
Illustration of the crack morphologies after sample failure for the reference sample CSH-60/40 and the calcium–aluminate-based cements #80/Silica and CAP#71/Silica.

**Figure 14 materials-17-01320-f014:**
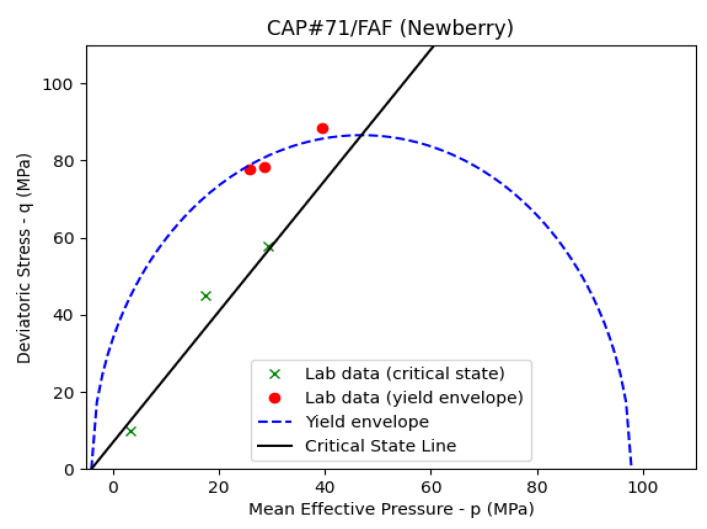
MCC model yield envelope and critical state line according to the lab experiments.

**Figure 15 materials-17-01320-f015:**
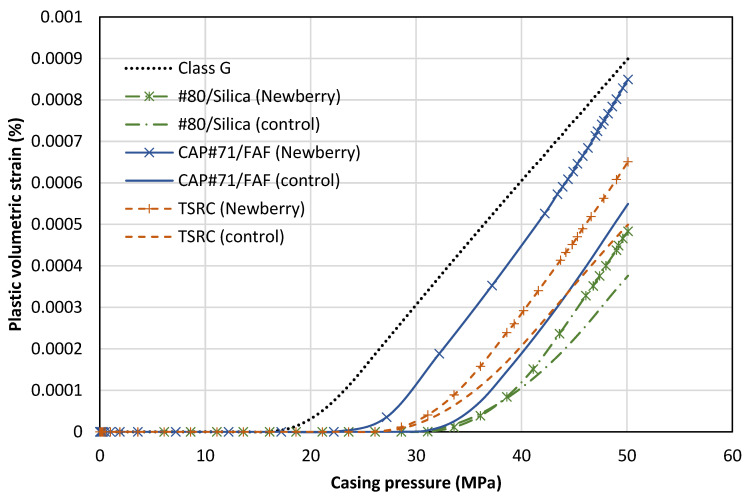
Average plastic volumetric strain of the cement elements in the numerical model versus casing pressure.

**Figure 16 materials-17-01320-f016:**
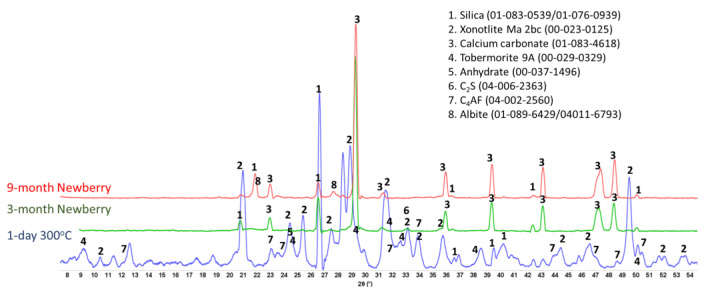
XRD patterns of the reference CSH-60/40 sample (after 1 day of autoclaving at 300 °C) and samples exposed in the Newberry well for 3 or 9 months.

**Figure 17 materials-17-01320-f017:**
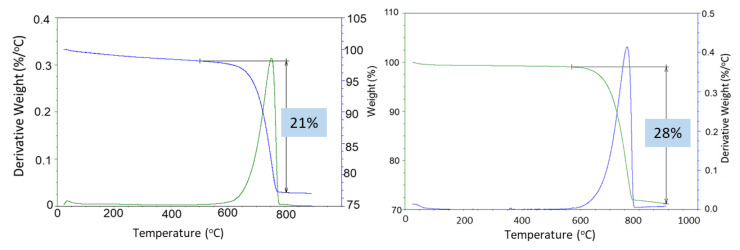
TGA/DTG curves of the CSH-60/40 samples exposed in the Newberry well for 3 months (**left**) or 9 months (**right**).

**Figure 18 materials-17-01320-f018:**
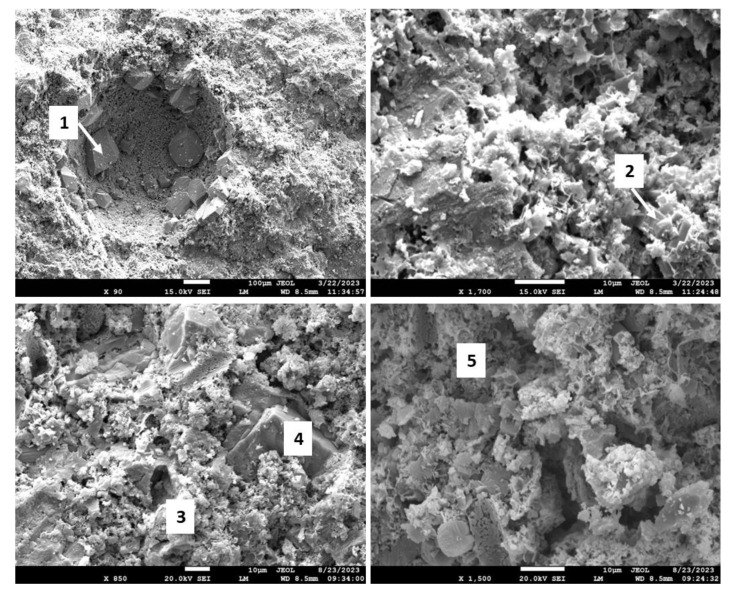
Photomicrographs of CSH-60/40 samples exposed in the Newberry well for 3 months (**top**) or 9 months (**bottom**).

**Figure 19 materials-17-01320-f019:**
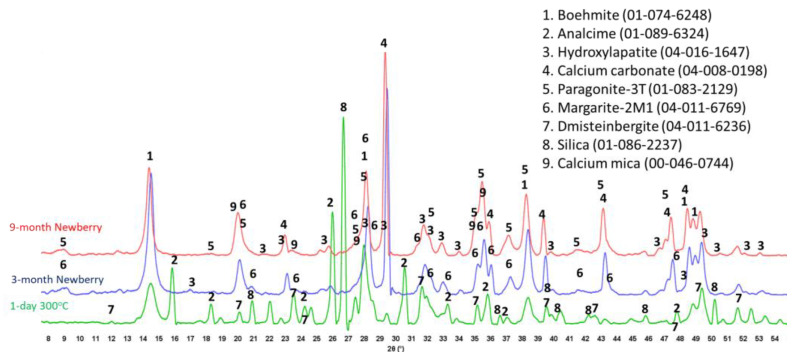
XRD patterns of the reference CAP#71/FAF sample (after 1 day of autoclaving at 300 °C) and samples exposed in the Newberry well for 3 or 9 months.

**Figure 20 materials-17-01320-f020:**
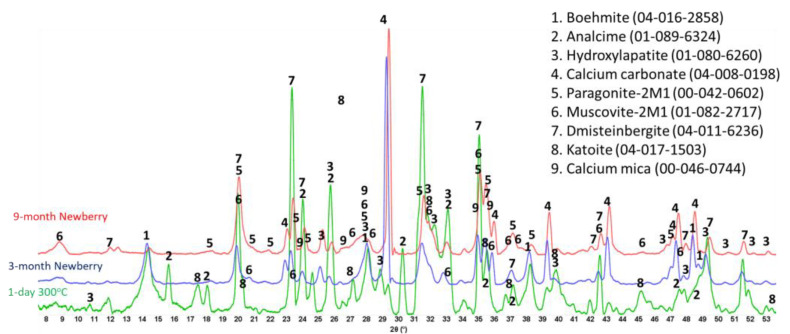
XRD patterns of the reference CAP#50/FAF sample (after 1 day of autoclaving at 300 °C) and samples exposed in the Newberry well for 3 or 9 months.

**Figure 21 materials-17-01320-f021:**
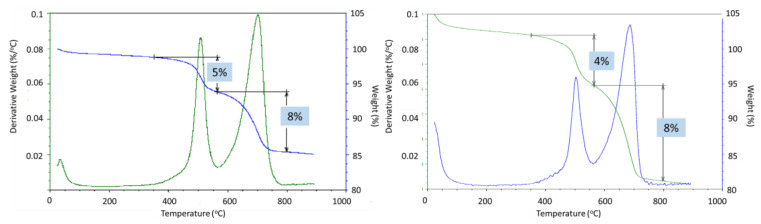
TGA/DTG curves of the CAP#71/FAF samples exposed in the Newberry well for 3 months (**left**) or 9 months (**right**).

**Figure 22 materials-17-01320-f022:**
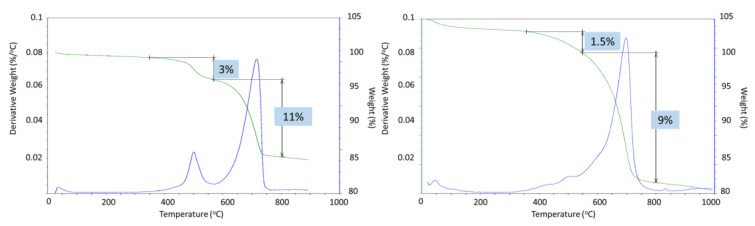
TGA/DTG curves of the CAP#50/FAF samples exposed in the Newberry well for 3 months (**left**) or 9 months (**right**).

**Figure 23 materials-17-01320-f023:**
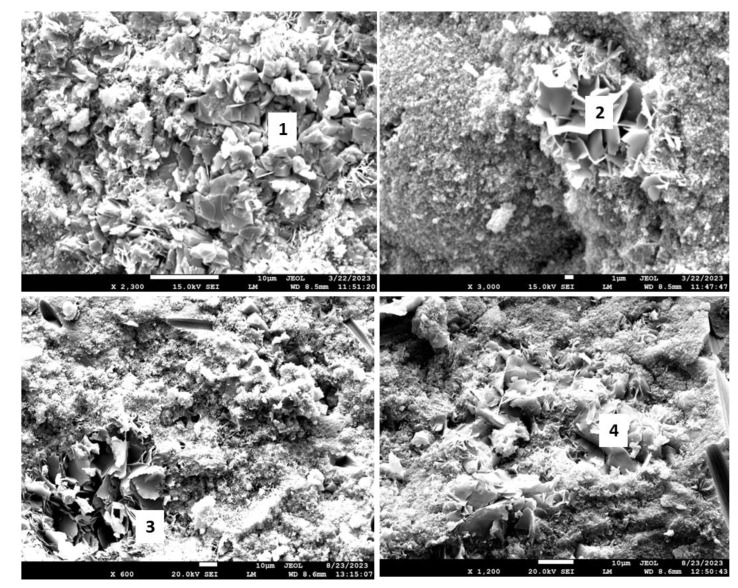
Photomicrographs of CAP#71/FAF samples exposed in the Newberry well for 3 months (**top**) or 9 months (**bottom**).

**Figure 24 materials-17-01320-f024:**
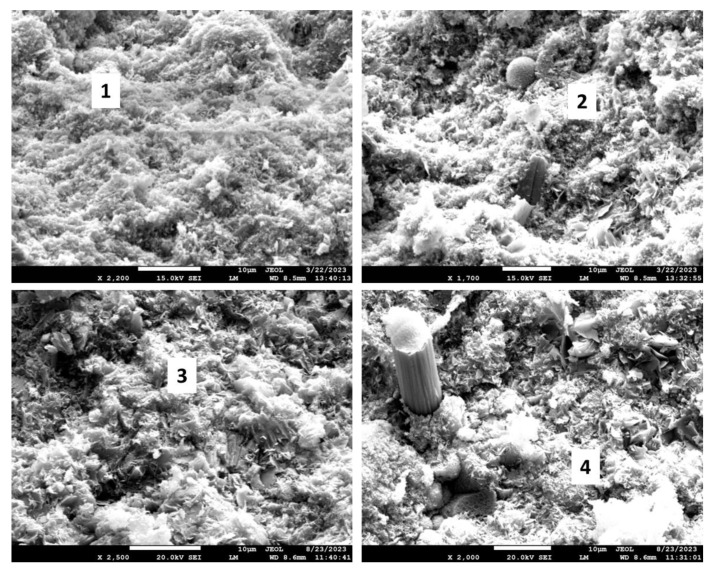
Photomicrographs of CAP#50/FAF samples exposed in the Newberry well for 3 months (**top**) or 9 months (**bottom**).

**Figure 25 materials-17-01320-f025:**
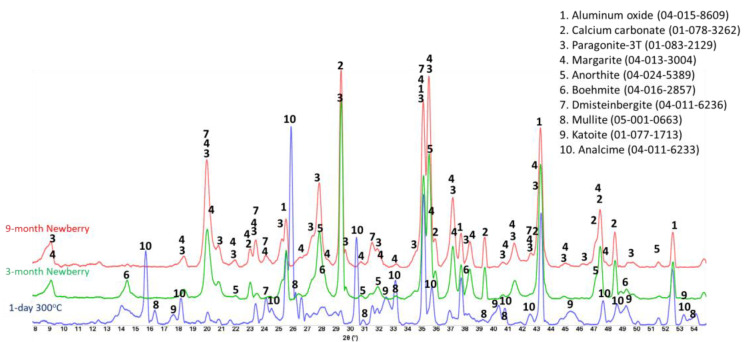
XRD patterns of the reference TSRC sample (after 1 day of autoclaving at 300 °C) and samples exposed in the Newberry well for 3 or 9 months.

**Figure 26 materials-17-01320-f026:**
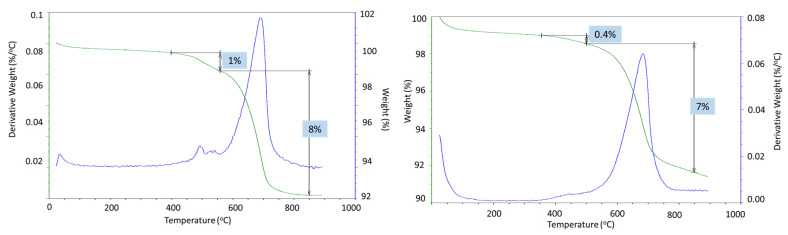
TGA/DTG curves of the TSRC samples exposed in the Newberry well for 3 months (**left**) or 9 months (**right**).

**Figure 27 materials-17-01320-f027:**
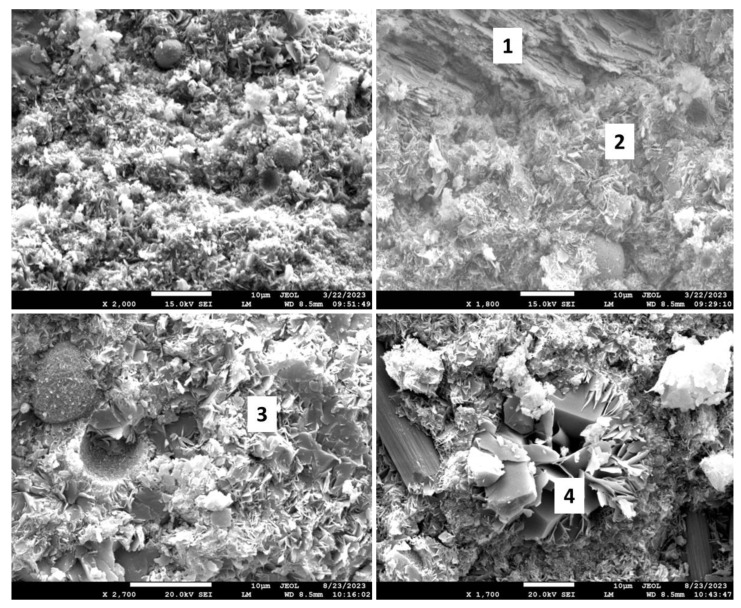
Photomicrographs of TSRC samples exposed in the Newberry well for 3 months (**top**) or 9 months (**bottom**).

**Figure 28 materials-17-01320-f028:**
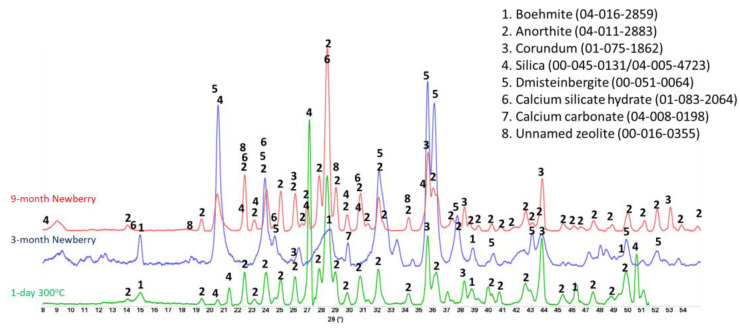
XRD patterns of the reference #80/Silica sample (after 1 day of autoclaving at 300 °C) and samples exposed in the Newberry well for 3 or 9 months.

**Figure 29 materials-17-01320-f029:**
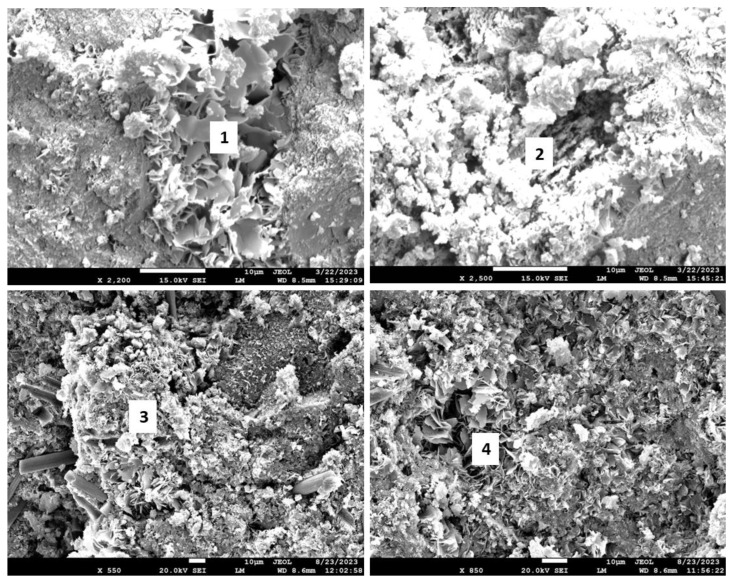
Photomicrographs of #80/Silica samples exposed in the Newberry well for 3 months (**top**) or 9 months (**bottom**).

**Table 1 materials-17-01320-t001:** Cement designs evaluated in this work.

Formulation	Composition (wt.%, Activator % by Weight of Cement Blend)
CSH-60/40	OPC/SiO_2_ (60/40)
TSRC	CAC#80/FAF/SMS (60/40, SMS at 6%)
CAP#71/FAF	CAC#71/FAF/SHMP (70/30, SHMP at 6%)
CAP#71/Silica	CAC#71/ SiO_2_/SHMP (70/30, SHMP at 6%)
CAP#71/Silica/MK	CAC#71/FAF/MK/SHMP (60/30/10, SHMP at 6%)
CAP#50/FAF	CAC#50/FAF/SHMP (70/30, SHMP at 6%)
#71/Silica	CAC#71/SiO_2_ (60/40)
#80/Silica	CAC#80/SiO_2_ (60/40)
NAS-M1	Gibbsite/SiO_2_/SMS (60/40, SMS and Zr at 5% each)

**Table 2 materials-17-01320-t002:** Summary of input parameters used in the numerical model.

Parameter	Unit	Casing	Cement	Rock
Inner diameter	m	0.0563	0.0603	0.0763
Outer diameter	m	0.603	0.0763	0.15
Cement bond strength	MPa	0.5	-	3
Young’s modulus	GPa	210	Varies by formulation	13.5
Poisson ratio	-	0.3	Varies by formulation	0.3
Mohr–Coulomb friction	deg	-	-	35
Mohr–Coulomb cohesion	MPa	-	-	13.3

**Table 3 materials-17-01320-t003:** Results from drained triaxial compression testing.

Formulation	*φ*	*σ* *con*	*p* *p*	*σ* *ax*	*σ* *res*	*E*	*ν*
	%	MPa	MPa	MPa	MPa	GPa	-
#80/Silica (Newberry)	0.43	5.0	2.5	46.3	26.3	14.4	0.14
	0.43	12.5	2.5	72.3	59.2	15.1	0.18
#80/Silica (control)	0.48	5.0	2.5	26.6	24.1	5.1	0.20
	0.48	12.5	2.5	40.2	40.0	5.3	0.13
CAP#71/FAF (Newberry)	0.38	5.0	2.5	83.4	50.0	19.3	0.18
	0.38	12.5	2.5	101.0	70.4	20.0	0.18
CAP#71/FAF (control)	0.40	5.0	2.5	85.1	39.0	17.3	0.17
	0.40	12.5	2.5	106.1	80.0	16.9	0.19
	0.40	17.5	2.5	109.8	92.0	14.6	0.18
TSRC (Newberry)	0.48	5.0	2.5	37.6	25.1	13.9	0.14
	0.48	5.0	2.5	40.3	25.4	13.6	0.26
	0.48	12.5	2.5	51.8	50.7	14.8	0.15
TSRC (control)	0.49	5.0	2.5	25.6	22.8	9.2	0.21
	0.49	12.5	2.5	39.9	39.9	8.1	0.15

**Table 4 materials-17-01320-t004:** Modified Cam-Clay parameters obtained for three formulations.

Formulation	*p_c_*	*p_t_*	*M*	*λ*	*κ*
	MPa	MPa	-	-	*-*
#80/Silica (Newberry)	58.74	5.13	1.6	0.0154	0.0035
#80/Silica (control)	28.56	3.45	1.3	0.0435	0.01002
CAP#71/FAF (Newberry)	97.85	4	1.7	0.0109	0.0025
CAP#71/FAF (control)	102.84	4	1.7	0.0131	0.003
TSRC (Newberry)	40.88	4.91	1.5	0.0149	0.0034
TSRC (control)	25.51	4.08	1.3	0.0248	0.0057

**Table 5 materials-17-01320-t005:** Elemental composition in the selected representative locations of the CSH-60/40 samples exposed in the Newberry well for 3 or 9 months (the locations of the analyses are shown in Figure 18).

Element	Location	Weight Percent (% Error)	Identified Phase	Location	Weight Percent (% Error)	Identified Phase
3-Month Exposure						
C	1	12.93 (0.45)	Calcium Carbonate (CaCO_3)_	2	12.08 (0.52)	Calcium Carbonate (CaCO_3_)
O		49.91 (1.20)			50.30 (0.90)	
Si		1.01 (0.10)			4.17 (0.14)	
Ca		35.07 (0.53)			31.11 (0.55)	
Fe		1.08 (0.29)			1.88 (0.31)	
9-Month Exposure						
C	3	9.65 (1.60)	Calcium bicarbonate (CaHCO_3_)	4	8.93 (2.38)	Silica (SiO_2_)
O		35.07 (0.91)			32.18 (1.03)	
Al		1.83 (0.14)			-	
Si		28.98 (0.63)			54.45 (1.52)	
Ca		19.28 (0.45)			2.63 (0.17)	
Fe		4.60 (0.30)			1.33 (0.27)	
C	5	11.94 (3.59)	Calcium bicarbonate (CaHCO_3_)			
O		26.00 (2.00)				
Al		2.44 (0.38)				
Si		24.43 (1.32)				
Ca		16.69 (0.99)				
Fe		18.50 (1.34)				

**Table 6 materials-17-01320-t006:** Elemental composition in the selected representative locations of the CAP#71/FAF samples exposed in the Newberry well for 3 or 9 months (the locations of the analyses are shown in Figure 23).

Element	Location	Weight Percent (% Error)	Identified Phase	Location	Weight Percent (% Error)	Identified Phase
3-Month Exposure						
C	1	11.7 (0.45)	Calcium Carbonate (CaCO_3)_	2	-	Dmisteinbergite (CaAl_2_Si_2_O_8_)
O		48.30 (1.2)			43.02 (0.90)	
Si		-			16.66 (0.40)	
Ca		38.01 (0.57)			9.24 (0.43)	
Al		1.26 (0.11)			18.99 (0.40)	
Fe		-			12.09 (0.93)	
9-Month Exposure						
O	3	30.05 (0.66)	Calcium mica (Al_3_Ca_0.5_Si_3_O_11_) or Margarite ((Na_0.2_Ca_0.8_Al_3.9_Si_2.1_)_11_(OH))	4	36.14 (0.53)	Margarite ((Na_0.2_Ca_0.8_Al_3.9_Si_2.1_)_11_(OH))
Na		1.22 (0.20)			2.01 (0.18)	
Al		27.85 (0.38)			27.26 (0.31)	
Si		26.16 (0.40)			25.69 (0.33)	
K		4.52 (0.20)			2.82 (0.15)	
Ca		7.64 (0.24)			4.69 (0.17)	
Fe		2.11 (0.35)			1.39 (0.23)	

**Table 7 materials-17-01320-t007:** Elemental composition in the selected representative locations of the CAP#50/FAF samples exposed in the Newberry well for 3 or 9 months (the locations of the analyses are shown in Figure 24).

Element	Location	Weight Percent (% Error)	Identified Phase	Location	Weight Percent (% Error)	Identified Phase
3-Month Exposure						
C	1	10.91 (0.65)	Multiple, amorphous matrix	2	-	Hydroxylapatite, aluminum-silicate matrix
O		56.90 (0.90)			45.68 (0.80)	
Na		0.89 (0.10)			0.77 (0.16)	
Al		14.81 (0.32)			23.53 (0.24)	
Si		9.57 (0.23)			12.12 (0.21)	
P		0.93 (0.09)			4.13 (0.17)	
Ca		4.75 (0.16)			12.39 (0.24)	
Fe		1.24 (0.20)			1.38 (0.32)	
9-Month Exposure						
O	3	38.34 (0.74)	Calcium–aluminum–silicate matrix	4	30.05 (0.59)	Margarite ((Na_0.2_Ca_0.8_Al_3.9_Si_2.1_)_11_(OH))
Na		1.31 (0.22)			1.19 (0.20)	
Al		19.26 (0.32)			23.89 (0.32)	
Si		19.28 (0.35)			23.46 (0.34)	
K		2.64 (0.15)			2.78 (0.15)	
Ca		15.92 (0.30)			4.15 (0.16)	

**Table 8 materials-17-01320-t008:** Elemental composition in the selected representative locations of the TSRC samples exposed in the Newberry well for 3 or 9 months (the locations of the analyses are shown in Figure 27).

Element	Location	Weight Percent (% Error)	Identified Phase	Location	Weight Percent (% Error)	Identified Phase
3-Month Exposure						
O	1	47.68 (0.52)	Boehmite (AlOOH)	2	48.08 (0.85)	Margarite (CaAl_4_Si_2_O_10_(OH)_2_)
Al		47.73 (0.41)			31.71 (0.28)	
Si		4.59 (0.31)			15.95 (0.24)	
Ca		-			4.26 (0.21)	
9-Month Exposure						
O	3	28.64 (0.75)	Anorthite, potassium bearing (like (Ca, Na) (Si, Al)_4_O_8_)	4	38.99 (0.62)	Dmisteinbergite (CaAl_2_Si_2_O_8_)
Na		-			1.61 (0.18)	
Al		25.62 (0.39)			19.23 (0.28)	
Si		30.11 (0.46)			14.56 (0.27)	
K		6.96 (0.25)			1.37 (0.13)	
Ca		6.81 (0.26)			23.16 (0.33)	
Fe		1.86 (0.39)			1.08 (0.26)	

**Table 9 materials-17-01320-t009:** Elemental composition in the selected representative locations of the #80/Silica samples exposed in the Newberry well for 3 or 9 months (the locations of the analyses are shown in Figure 29).

Element	Location	Weight Percent (% Error)	Identified Phase	Location	Weight Percent (% Error)	Identified Phase
3-Month Exposure						
O	1	46.84 (0.9)	Dmisteinbergite (CaAl_2_Si_2_O_8_)	2	43.98 (0.05)	Boehmite (AlOOH)
Al		22.02 (0.52)			40.21 (0.35)	
Si		20.03 (0.59)			2.63 (0.25)	
Ca		11.11 (0.59)			10.34 (0.29)	
9-Month Exposure						
O	3	29.48 (1.02)	Anorthite, potassium bearing (like (Ca, Na) (Si, Al)_4_O_8_)	4	31.11 (1.17)	Unnamed zeolite (K_2.84_Ca_1.43_Al_5.7_Si_10.3_O_32_ 10.6H_2_O)
Na		1.17 (0.29)			1.33 (0.33)	
Al		23.82 (0.49)			21.30 (0.54)	
Si		25.54 (0.56)			23.68 (0.62)	
K		4.30 (0.31)			10.48 (0.45)	
Ca		15.68 (0.46)			12.10 (0.52)	

## Data Availability

Data are contained within the article.

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
