# Peer review of "Assessment of Cementitious Composites for High-Temperature Geothermal Wells"

_materials, 2024, doi:10.3390/ma17061320_

Round 1

Reviewer 1 Report

Comments and Suggestions for Authors

The study is systematic and all experimental aspects are described for best reproducability.

My only remark is to revise the XRD pattern. Intensity scale of reflexes in cps should be added, it should be clearly stated whether the background was substarcted and all missing minor reflexes should also be indexed. For better comparability, auxiliary lines along the 2theata axis should be added.

Author Response

We thank the reviewer for such a positive review of the paper. We added a sentence on the subtraction of the background from the XRD patterns to the section on materials analyses. 

However, we prefer no to add the cps axes since no phase quantification is reported and the overlaying of the patterns results in arbitrary cps. The statements made in the paper refer only to qualitative changes in the patterns, which are obvious in the case of OPC/silica formulation with a complete transformation of all the original peaks to those of calcium carbonate. 

For other formulations statements on changes in the peaks intensity are made only where peaks nearly disappear or appear in the XRD patterns. We reviewed the statements that we made accordingly. 

Reviewer 2 Report

Comments and Suggestions for Authors

This manuscript presents a thorough investigation into the properties of cementitious composites under high temperatures. However, there are several points that require attention before publication:

  1. The Introduction section is too long. Please consider condensing it to include only closely relevant information.
  2. Clearly demonstrate the motivation behind studying the three cement formulations and the necessity of incorporating carbon microfibers in the research. Provide references to support these choices.
  3. Identify the existing research gap and elaborate on how this study addresses and fills that gap. Clearly define the contribution of this research.
  4. Section 3.1 addresses unconfined mechanical properties and sample water-fillable porosity. Please include the impact of sample water-fillable porosity on unconfined mechanical properties.
  5. Regarding Fig. 18, clarify the authors' rationale for identifying location 1 as calcium carbonate crystals. The reviewer suggests including an element mapping along with micrographs and element composition for better clarification.
  6. The contribution of this study needs further clarification. Specify how the findings will impact future research in this field and provide insights into potential avenues for future work.
Comments on the Quality of English Language

N/A

Author Response

1. The introduction is long since it covers several important points: 1) importance and potential of HT geothermal energy for green energy production; 2) difficult HT geothermal environments for materials to survive and a wide range of conditions that materials experience in such geothermal wells; 3) significant body of research done under low T and short exposure times with the conclusions that are not predictive of cement survival under the real HT geothermal well conditions; 4) description of earlier research on cement formulations specifically designed and tested for applications in HT geothermal wells, also tested in this work ; 5) brief description of relevant laboratory setups for cement evaluation also used in the current study and earlier modeling efforts on integrity of cement sheath; 6) statement on what the current paper is about.

To shorten the introduction we removed the first paragraph and shortened one of the paragraphs. We also added a statement on the knowledge gap that this paper is addressing to the introduction.

2. We added explanation on the use of MCF and a citation under the "Materials"

3. A statement to this effect is added to the introduction and a paragraph is added to the conclusions.

4. There is no simple correlation between mechanical properties and water-fillable porosity. For examples, although it is tempting to state that low-porosity of calcium phosphate cements results in their high strength, the porosity is not the only factor that affects mechanical properties. This is clearly visible in the case of OPC/silica that lost its mechanical properties while keeping low porosity. The effect of porosity on other properties such as toughness and moduli are even more complex. Nevertheless, we added a sentence about the highest and lowest strength correlating with the lowest and highest porosity of tested formulations. 

5. The location 1 was identify as calcium carbonate based on the elemental mass % given in the table below the photomicrographs and morphology of the crystals. We do not believe that addition of mapping will bring any more information than already given but it will complicate the paper that is already quite complex.

6. A paragraph was added to the conclusions to clarify contribution of this study to geothermal cement research.

Reviewer 3 Report

Comments and Suggestions for Authors

The manuscript „Assessment of Cementitious Composites for High Temperature Geothermal Wells“ is interesting, and the theme is novel.

The authors studied high-temperature geothermal wells that can provide green power 24 hours a day, 7 days a week. In harsh environmental and operational conditions, the long-term durability requirements of such wells require special cementitious composites for well construction. This paper reports a comprehensive assessment of geothermal cement composites in cyclic pressure function laboratory tests and field exposures in an HT geothermal well (300–350 °C) as well as a numerical model to complement the experimental results. Performances of calcium-aluminate-cement (CAC)-based composites and calcium-free cement were compared against the reference Ordinary Portland Cement (OPC)/silica blend.

The research is scientifically sound. Not too many researchers are studying geothermal wells, which makes this study stand out. The work is nicely conceptualized. The methodology is clear. The text is readable. Figures and tables are of adequate quality and placed in the right positions in the text. There is no error, fact, or logic in the presentation of the results. The discussion follows the presented results adequately. Conclusions are appropriate. References are up-to-date.

The manuscript can be accepted after the correction of several minor mistakes. The list is provided below:

Key words: can they be shorter and more specific?

The introduction is elaborate. It includes all of the main references. However, could authors add a sentence or two about how their work is closing the scientific gap in knowledge?

Materials: Please provide chemical compositions for utilized cements and XRD diffractograms. The same goes for other applied raw materials. If authors think that adding these tables and graphs to the Materials Section would overweight the manuscript, they can be added as supplementary materials. Either way, in my opinion, they are necessary for further discussion.

The data provided in Table 1 is not very clear. Can the mix designs be provided?

Author Response

We thank the reviewer for the positive review of the paper.

  1. We changed the key words
  2. We modified the introduction and stated the knowledge gap addressed by this research
  3. The XRD compositions of the used materials are given under "materials". These are closely controlled cements of high quality from known producers. Their compositions do not change significantly from batch to batch. We believe that provided information is sufficient.
  4. We added an explanation to the formulations reported in Table 1 in the text of the paper with an example explaining how the information is presented in that table.